## OPEN

# Label-free nanofluidic scattering microscopy of size and mass of single diffusing molecules and nanoparticles

Barbora Špačková [1 ✉], Henrik Klein Moberg[1], Joachim Fritzsche [1], Johan Tenghamn[1], Gustaf Sjösten[2], Hana Šípová-Jungová [1], David Albinsson[1], Quentin Lubart[3], Daniel van Leeuwen[3], Fredrik Westerlund[3], Daniel Midtvedt[2], Elin K. Esbjörner [3], Mikael Käll [1], Giovanni Volpe [2] and Christoph Langhammer [1 ✉]

Label-free characterization of single biomolecules aims to complement fluorescence microscopy in situations where labeling compromises data interpretation, is technically challenging or even impossible. However, existing methods require the investigated species to bind to a surface to be visible, thereby leaving a large fraction of analytes undetected. Here, we present nanofluidic scattering microscopy (NSM), which overcomes these limitations by enabling label-free, real-time imaging of single biomolecules diffusing inside a nanofluidic channel. NSM facilitates accurate determination of molecular weight from the measured optical contrast and of the hydrodynamic radius from the measured diffusivity, from which information about the conformational state can be inferred. Furthermore, we demonstrate its applicability to the analysis of a complex biofluid, using conditioned cell culture medium containing extracellular vesicles as an example. We foresee the application of NSM to monitor conformational changes, aggregation and interactions of single biomolecules, and to analyze single-cell secretomes.

Fluorescence microscopy has been a long-standing workhorse in biochemistry and biophysics[1–5]. However, a key limitation is the need for labeling with fluorescent tags. Therefore, one current research frontier is the development of methods that enable label-free studies of single biological nanoparticles (BNPs) and biomolecules to complement fluorescence-based techniques. Label-free methods bypass the following limitations of fluorescence microscopy: (1) attaching a fluorescent label to a target may alter its properties[6,7]; (2) there is a limited number of label colors that can be used simultaneously; and (3) long-term measurements are complicated by photobleaching. In addition, in cell-secretion-related studies[8], it is very difficult to specifically label biomolecules of interest and, most importantly, it is only possible to detect predefined labeled secreted entities, which means unlabeled but potentially important entities are probably overlooked.

Label-free single-biomolecule detection has been enabled recently by dielectric microresonators[9], plasmonic approaches based on metallic continuous films[10] and nanostructures[11], and interferometric scattering microscopy (iSCAT)[12–16]. iSCAT has been used to investigate, for example, single cell secretion dynamics[13] and protein motion on a dielectric substrate[14], on a supported lipid bilayer[17,18] or on actin filaments[15]. Furthermore, in contrast to alternative methods, iSCAT enables quantitative molecular weight (MW) measurements of biomolecules, since the signal intensity is proportional to the weight of the imaged object[12,16]. However, and as a key point, these three label-free optical single-molecule detection methods require the investigated species to bind to a surface to be "visible". This requirement can lead to misinterpretations when applied to biomolecular interaction studies because binding

to a surface may affect the properties and accessibility of molecular binding sites[19]. In addition, binding is a selective process, meaning that only molecules that actually bind are detected, while a large fraction remains unseen. Consequently, the ability to label-free image and track diffusing single biomolecules directly in solution would constitute an important step in the field. However, the resolution of current state-of-the-art microscopy techniques allows imaging only of much larger diffusing objects, such as viruses[20], extracellular vesicles[21] (EVs), or dielectric particles[22]. This is mainly because the scattering cross-section of individual biomolecules is very small, preventing their direct detection, and because their fast Brownian motion permits the accumulation of the light they scatter only for the extremely short time they spend in a diffraction limited spot and/or before they diffuse out of the focal plane.

To overcome these limitations, we present NSM, which enables the real-time label-free imaging of single BNPs and biomolecules in solution down to the few tens of kilo-Daltons regime inside a nanofluidic channel, without the need for surface immobilization. Furthermore, it allows the simultaneous determination of MW from the optical contrast of the imaged nano-object, and of its hydrodynamic radius ($R_s$) and/or conformational state from the measured diffusivity.

## Results

**Nanofluidic scattering microscopy.** NSM works by imaging nanofluidic channels nanofabricated into an optically transparent matrix, such as $SiO_2$, by dark-field light-scattering microscopy (Fig. 1a). The channel cross-sectional dimensions can range from tens to hundreds of nanometers, depending on the size of the investigated

[1]Department of Physics, Chalmers University of Technology, Göteborg, Sweden. [2]Department of Physics, University of Gothenburg, Göteborg, Sweden. [3]Department of Biology and Biological Engineering, Chalmers University of Technology, Göteborg, Sweden. ✉e-mail: spackova@chalmers.se; clangham@chalmers.se

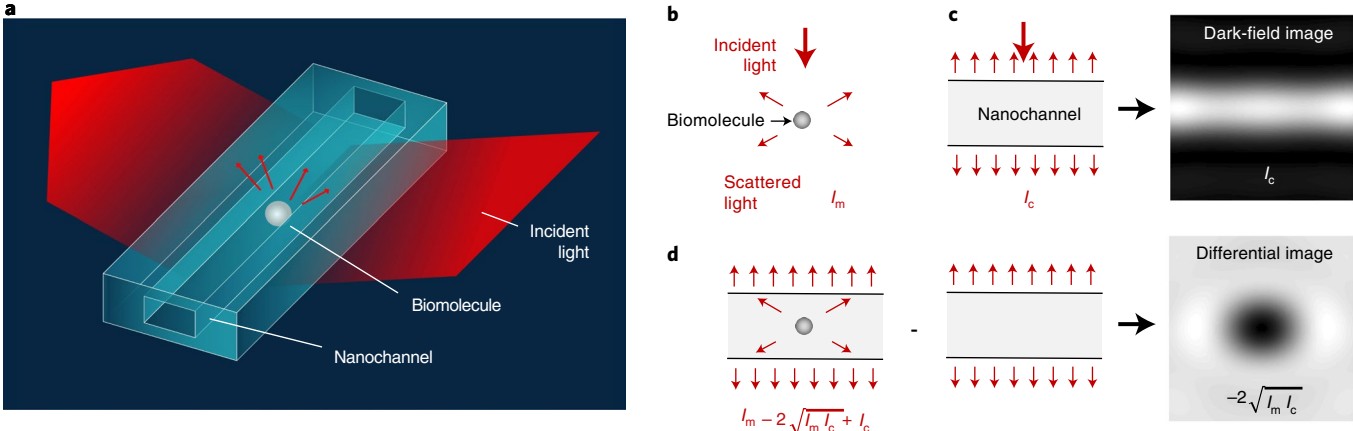

**Fig. 1 | Principle of NSM. a**, Artist's rendition of the experimental configuration where visible light irradiates a nanochannel with a biomolecule inside, and where the light scattered from the system is collected in dark-field configuration. **b**, Schematic of light scattered by a single biomolecule. **c**, Schematic of light scattered by a nanochannel and the corresponding dark-field image. **d**, Schematic of light scattered by a nanochannel with a single biomolecule inside, and the corresponding differential dark-field image obtained by subtracting an image of the empty nanochannel from the image of the nanochannel with the biomolecule inside.

nano-object, while their lengths typically exceed the microscope's field of view (Extended Data Fig. 1 and 2; see 'Nanofluidic chips' and 'Experimental setup' in Methods). In this arrangement, the nanochannels ensure the localization of the nano-objects within the microscope focal plane throughout the entire imaging process, similar to other tether-free microscopy methods[23]. Most importantly, the nanochannels improve the optical contrast of the imaged nano-object by several orders of magnitude.

To introduce the underlying principle, we consider a single biomolecule diffusing inside a nanochannel (Fig. 1a). The biomolecule and the nanochannel scatter light coherently into the collection optics, resulting in a total scattering intensity, $I_t = cI_0L|\alpha_t|^2k^3/4$, which depends on the incident intensity ($I_0$), the wavenumber of the light ($k$), the length of the illuminated part of the nanochannel ($L$), the collection efficiency ($c$) and the optical properties of the molecule and the nanochannel, defined by the total electric dipole polarizability ($\alpha_t$)[24]. The $\alpha_t$ can be determined by the respective electric dipole polarizabilities of the biomolecule, $\alpha_m$, and of the nanochannel, $\alpha_c$, and can be approximated as $\alpha_t \approx \alpha_c + \alpha_m/L^2$, where the length $L$ stems from the fact that the polarizability of a nanochannel is two-dimensional, whereas the polarizability of a molecule is three-dimensional (see Section 1 in Supplementary Information). The total scattering intensity collected from a region of a nanochannel of length $L = 3\pi/(2k)$ that contains a biomolecule can then be written as $I_t \approx I_c + I_m - 2\sqrt{I_cI_m}$, where $I_c = cI_0L|\alpha_c|^2k^3/4$ and $I_m = cI_0|\alpha_m|^2k^4/(6\pi)$ are the scattering intensities produced by the biomolecule (Fig. 1b) and the nanochannel (Fig. 1c) in isolation, respectively. The minus sign in front of the interference term stems from the fact that the nanochannel resides in a high-refractive-index background ($\alpha_c < 0$ since $n_{H_2O} \approx 1.33 < n_{SiO_2} \approx 1.46$), while the opposite is true for the biomolecule ($\alpha_m > 0$ since $n_m > n_{H_2O}$). Since scattering from subwavelength objects scales as volume squared, and because the nanochannel volume is much larger than the molecular volume, $I_c \gg I_m$, the scattering signal stemming from the biomolecule is not directly detectable against the nanochannel background. However, by subtracting the scattering image produced by an empty nanochannel ($I_c$) from a scattering image of a nanochannel with a biomolecule inside ($I_t$), the presence of the biomolecule can be revealed through the interference term, which contributes a sizeable (negative) signal $\Delta I_t = I_t - I_c \approx -2\sqrt{I_cI_m}$ (Fig. 1d) that can be several orders of magnitude larger than the scattering intensity produced by the

biomolecule alone ($I_m$) outside the nanochannel. This key feature of NSM, which is related closely to homodyne detection in laser interferometry[16,20], thus enables the direct imaging of diffusing biomolecules and other nano-objects inside a nanochannel.

**Experimental setup and data analysis.** To approach the photon shot noise level necessary for NSM imaging, we used a dark-field microscope (Mad City Labs RM21), a polychromatic light source (NKT Photonics, SuperK EXTREME EXB-6) and a high-speed CMOS camera (Andor, Zyla), enabling the recording of movies at 200 frames per second at an averaged noise level of 0.005%. This is low enough to distinguish the optical contrast generated by single biomolecules with MWs ranging down to tens of kilo-Daltons. At the same time, the temporal resolution is high enough to capture their fast Brownian motion (Fig. 2). To extract single-molecule information from the raw data, every frame of a movie is corrected for long- and short-term mechanical and intensity instabilities, and an empty nanochannel background is estimated from every frame and subtracted from the signal. The resulting time-series of differential dark-field images is normalized by the intensity profile of an empty nanochannel to correct for inhomogeneous illumination (see Section 5 in Supplementary Information). A selection of frames from a video so obtained of a single thyroglobulin protein (669 kDa) diffusing inside a nanochannel is shown in Fig. 2a (Supplementary Movie 1). Plotting the entire time-sequence of obtained images in a corresponding kymograph displays the movement of the protein in the nanochannel, where each horizontal line corresponds to the optical signal averaged across the short axis of the nanochannel (Fig. 2b). A corresponding library of kymographs of different types of protein and double-stranded DNA molecules is displayed in Fig. 2c–h, obtained for the proteins inside a nanochannel with cross-sectional area $A_I = 100 \times 27$ nm² (Channel I in Extended Data Fig. 3a) and for the DNA molecules inside a nanochannel with cross-sectional area $A_{II} = 110 \times 72$ nm² (Channel II in Extended Data Fig. 3b).

**MW and $R_s$ determination.** NSM can not only image individual biomolecules, but also determine their MW and hydrodynamic properties. The hydrodynamic properties contain important information about $R_s$ and/or shape. The MW determination is enabled by the integrated optical contrast (iOC) being linearly dependent on the polarizability of a biomolecule, $\alpha_m$, which is linearly proportional

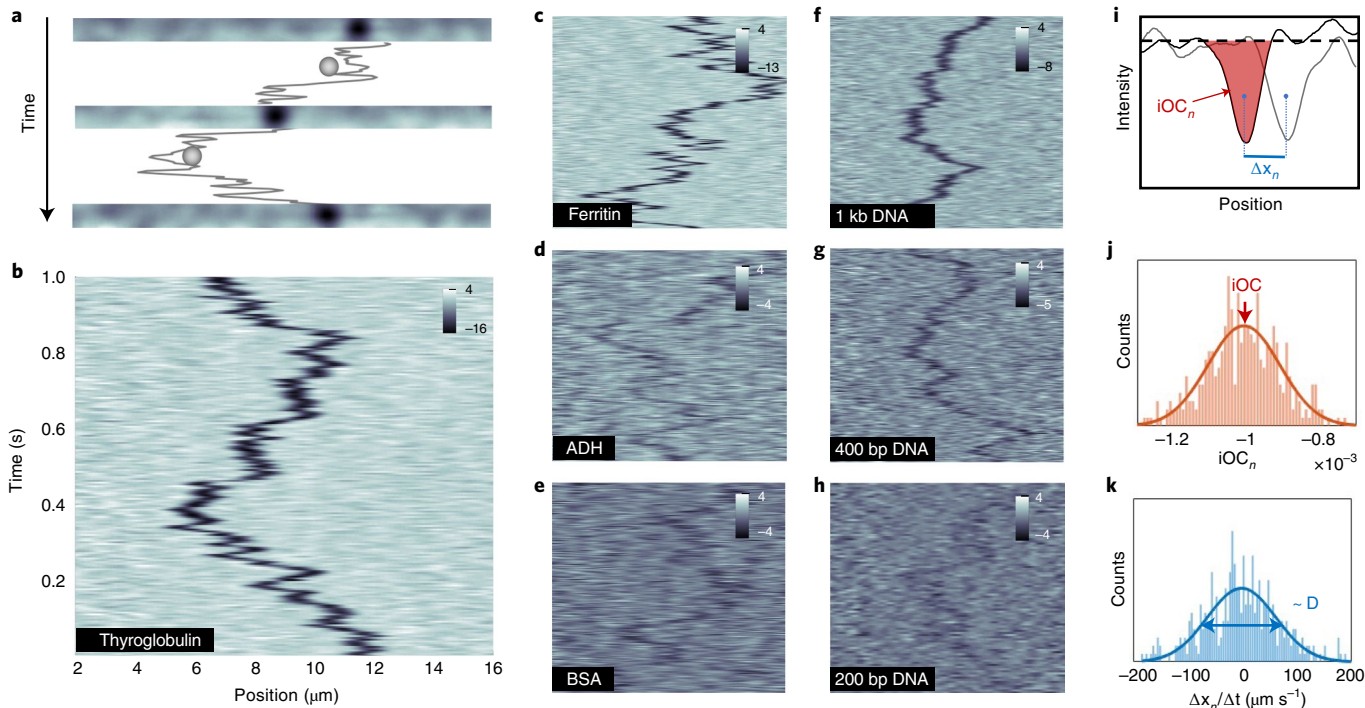

**Fig. 2 | Time-resolved NSM of diffusing single biomolecules. a**, Selection of differential images of a nanochannel containing a diffusing single thyroglobulin, MW = 669 kDa. Its trajectory is depicted between the images. **b–h**, Kymographs of single proteins inside Channel I with $A_I = 100 \times 27$ nm² and DNA molecules inside Channel II with $A_{II} = 110 \times 72$ nm²: thyroglobulin (669 kDa) (**b**), ferritin (440 kDa) (**c**), ADH (150 kDa) (**d**), BSA, (66 kDa) (**e**), 1 kb DNA (650 kDa) (**f**), 400 bp DNA (264 kDa) (**g**), 200 bp DNA (132 kDa) (**h**). The space and time coordinates of **c–h** correspond to those shown in **b**. Contrast is expressed in relative units divided by 10⁻⁴ (**i–k**). Principle of iOC and D evaluation. **i**, $iOC_n$ and space displacement ($\Delta x_n$) are determined from the nth and (n+1)th frame of the kymograph, respectively. **j** Histogram of $iOC_n$ and **k** space displacement in time ($\Delta x_n / \Delta t$) evaluated in each time step of a trajectory of a single thyroglobulin.

to its MW[12,16] as $\alpha_m \cong a \cdot$ MW, where $a = 0.46$ Å³·Da⁻¹ corresponds to values extracted from both measurements[25] and calculations[16] of optical properties of a large number of proteins (see Section 3 in Supplementary Information). We also note that the linear dependency of $\alpha_m$ on MW is connected directly to the *f*-sum rule that relates the number of electrons in a system to its optical extinction cross-cross section[26]. iOC is also inversely proportional to the cross-sectional area of the nanochannel, $A$, and proportional to $\bar{n} = \left(1.5 n_{H_2O}^2 + 0.5 n_{SiO_2}^2\right) / \left(n_{H_2O}^2 - n_{SiO_2}^2\right)$ (see Section 1 in Supplementary Information). Both these factors are constant during the measurement and can be determined before it. Therefore, MW can be determined from iOC, as

$$\text{MW} = \text{iOC} \cdot \frac{A}{\bar{n}a}. \tag{1}$$

The diffusivity of a molecule can be obtained from the statistical analysis of its movement[27]. Subsequently, by approximating it as a hard neutral sphere, its hydrodynamic (Stokes) radius, $R_s$, can be estimated using the Stokes–Einstein equation corrected for hindrance effects associated with the diffusion of small objects in a restricted volume as[28]

$$R_s = K \cdot \frac{k_B T}{6\pi \eta D}, \tag{2}$$

where $k_B$ is the Bolzmann constant, $T$ is temperature, $\eta$ is the viscosity of the liquid in the nanochannel and $K$ is the hindrance factor that takes particle-wall hydrodynamic interactions and steric restrictions inside a nanochannel into account. It is dependent on the size of the nanochannel relative to the dimensions of the

biomolecule, and can be estimated using a phenomenological model suggested by Dechadilok et al.[28] as

$$K = \left(1 + 9\lambda \cdot \ln\lambda/8 - 1.56\lambda + 0.53\lambda^2 + 1.92\lambda^3 - 2.81\lambda^4 \right.$$
$$\left. + 0.27\lambda^5 + 1.1\lambda^6 - 0.44\lambda^7\right) / (1 - \lambda)^2,$$

where $\lambda = R_s/r$, $r = \sqrt{A/\pi}$ is the radius of a circle defined by an area $A$. We note here that the diffusivity in principle also can be affected slightly by other surface-related effects that are not included here, such as the partial-slip boundary condition[29].

To extract iOC and the position of the biomolecule along the nanochannel, $x$, we used a particle-tracking algorithm (see Section 6 in Supplementary Information). It evaluates each frame in the kymograph (Fig. 2i), finds the responses corresponding to a biomolecule and connects them in a trajectory. Each biomolecule is then represented by $N$ values of $iOC_n$ (Fig. 2j) and $N-1$ values of spatial displacement in time $\Delta x_n$ (Fig. 2k), where $N$ is the number of frames that constitute a single biomolecule trajectory. iOC pertaining to a single biomolecule is <u>defined</u> as mean value of $iOC_n$ and its $D$ can be calculated as $D = \overline{(\Delta x_n)^2}/2\Delta t + \overline{\Delta x_n \Delta x_{n+1}}/\Delta t$ [27]. Furthermore, we have corroborated these results with an independent analysis employing machine learning (ML) algorithms (see Section 7 in Supplementary Information) to derive iOC and $D$ directly from the raw data, whose results are in very good agreement with those of the standard analysis (SA) above.

As final comment, we note that the 28 nM molecular concentration chosen for the experiments is high enough to ensure sufficient throughput and, at the same time, low enough to enable correct and precise discrimination of individual biomolecules. It corresponds to 0.7 and 2 biomolecules on average per field of view in Channels I and

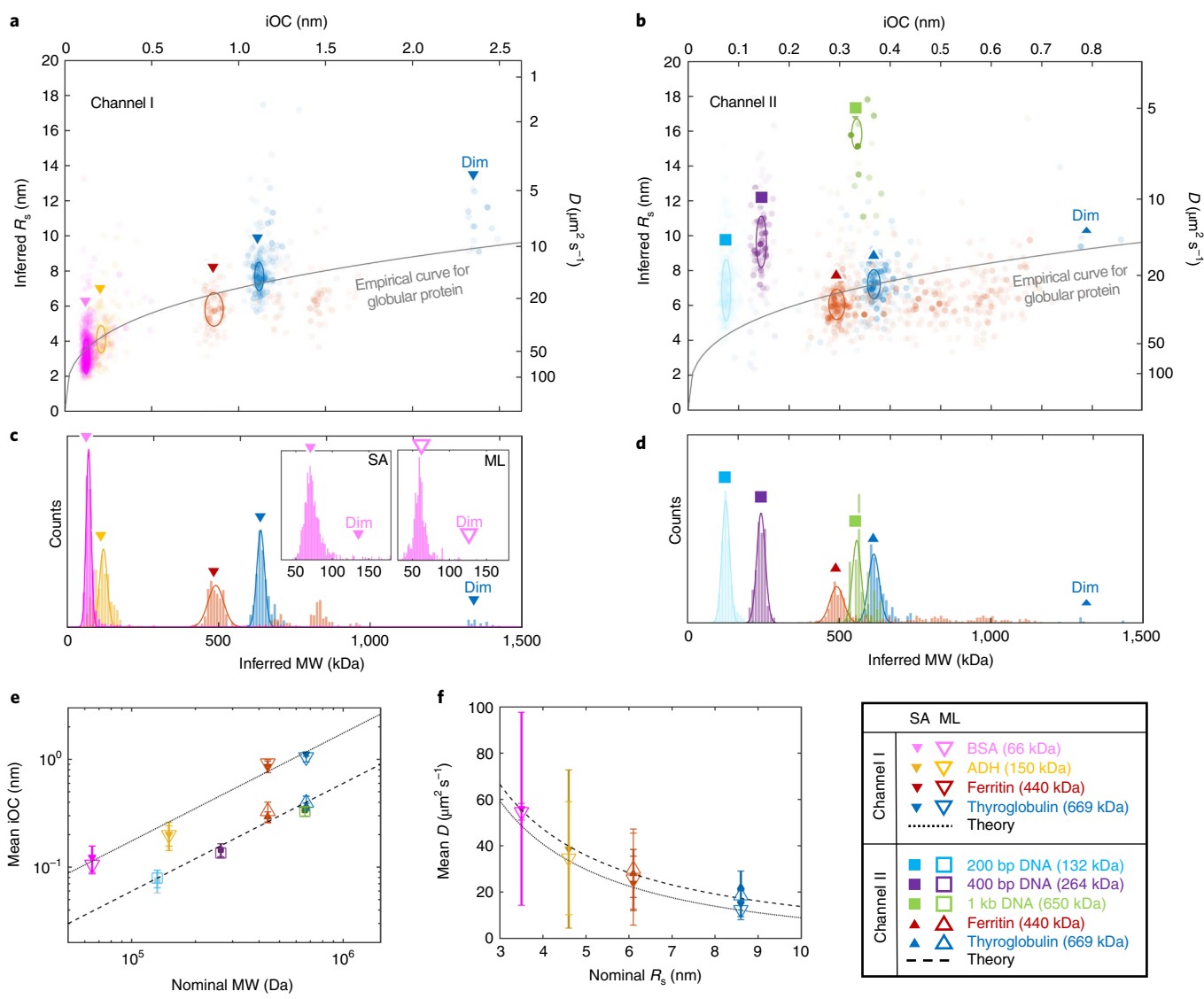

**Fig. 3 | A single-biomolecule library. a,b**, Scatter plots of iOC translated into MW using Eq. 1, and $D$ translated into $R_s$ using Eq. 2 for individual biomolecules of different types, measured in Channel I (**a**) and Channel II (**b**) and analyzed using SA. Each dot is extracted from a single biomolecule trajectory. Color intensity scales linearly with frame number of the trajectory ($N$). The highest intensity corresponds to $N = 1,100$ frames. The population of molecular monomers are marked by ellipses whose centers correspond to the $\overline{iOC}$ and $\bar{D}$, and their horizontal and vertical diameters to the resolution in iOC and $D$, respectively. The gray line corresponds to an empirical relationship between MW and $D$ for globular proteins[30]. **c,d**, MW histograms of the biomolecules in **a** and **b** translated from the iOC using Eq. 1. The insets in **c** show zoomed-in BSA histograms obtained by SA and ML, revealing a small dimer population (Dim). **e**, Dependency of $\overline{iOC}$ of protein monomers on nominal MW compared with the theoretical model (Eq. 1). **f**, Protein monomer $\bar{D}$ dependency on nominal $R_s$ compared with the confinement-corrected Stokes–Einstein equation (Eq. 2). **e** and **f** show the agreement between the independent results of SA and ML; data are presented as mean values and error bars correspond to the resolution in iOC and $D$, respectively; the presented values were derived from $n = 18$–695 trajectories (for $n$ specific for each measurement, see Source Data).

II, respectively. Lower concentrations can be studied by applying a flow rather than relying on diffusion alone to improve throughput. For higher concentrations, more advanced particle trackers are in development.

**A single biomolecule library.** Having established the theoretical framework of NSM, as well as its two main readout parameters, MW (calculated from iOC using Eq. 1) and $R_s$ (calculated from $D$ using Eq. 2), we now apply it to a library of proteins and DNAs, and use the two nanochannels with the different cross-sections introduced above, that is, Channel I and Channel II. Accordingly, each datapoint corresponds to iOC and $D$ determined from a trajectory of a

single biomolecule in Channel I (Fig. 3a) and Channel II (Fig. 3b). Each molecular species exhibits distinct populations and there is a clear correlation between $R_s$, MW and the shape of the biomolecules. Specifically, for proteins that are globular in shape, the $R_s$ scales approximately as $R_s \cong b \cdot MW^{1/3}$, where $b = 0.88$ nm Da$^{-1}$ was determined empirically[30] (gray solid lines in Fig. 3a,b). For DNA molecules that have a distinctly different geometry compared with globular proteins, $D$ is distinctly lower at the corresponding MW (Fig. 3b), as expected for elongated molecules[31].

To further analyze the obtained single biomolecule data, we plot one-dimensional histograms of iOC converted to MW for all biomolecules in Channel I and II obtained by SA (Fig. 3c,d).

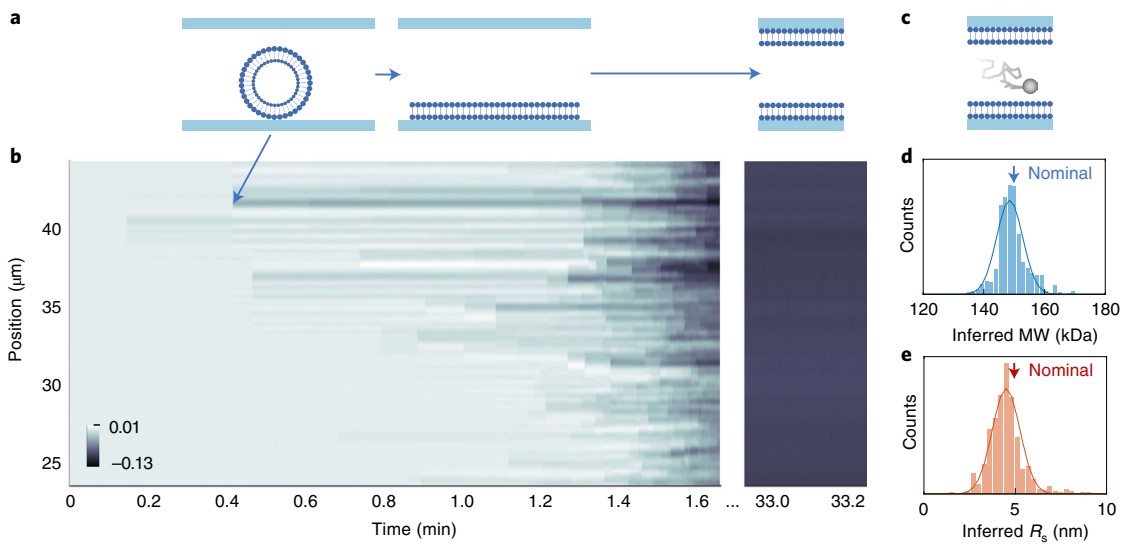

**Fig. 4 | Surface passivation. a**, Schematic of SLB formation. LUVs flow through the nanochannel, adsorb on the nanochannel wall, rupture and create patches of lipids that eventually connect and create a homogenous layer. **b**, Kymograph capturing the SLB formation in Channel V, as manifested by decrease in scattering intensity. **c**, Schematic of a biomolecule diffusing inside a nanochannel coated with an SLB. **d,e**, Inferred molecular weight (**d**) and $R_s$ (**e**) of the positively charged protein aldolase measured in Channel VI and analyzed by ML. The arrows indicate the nominal values (MW = 158 kDa, $R_s$ = 4.6 nm (ref. [32])).

Each detected trajectory is presented by $N/\sum N$ counts of its determined iOC where $\sum N$ is the sum of the number of frames of all the trajectories that were identified in the sample. This way, the numbers of counts within the peaks correspond to the (relative) concentration of different populations present in the sample. To evaluate the main peaks that correspond to molecular monomers, we fit the histograms with Gaussian distributions using the Matlab curve fitting toolbox. The mean value of iOC ($\overline{iOC}$) is thus defined as the center of the Gaussian peaks, while the resolution in iOC is defined by their full-width-at-half-maximum (FWHM). For Channel I this translates into a MW resolution of 20–30 kDa and for Channel II of 30–40 kDa (details in Section 10 in Supplementary Information), which defines the limits for resolving different populations in a sample. The resolution reached for thyroglobulin in Channel I (30 kDa) corresponds to a threefold improvement compared with iSCAT, which determines MW from single binding/unbinding events[16]. This improvement is enabled by the ability of NSM to track the diffusing biomolecules during the entire time they spend in the field of view, which, in some cases, can be much longer than the residence time on a surface.

As the final analysis step, we investigate the dependency of experimentally determined $\overline{iOC}$ on nominal MWs of monomers (see 'Biomolecular solutions' in Methods), compared with Eq. 1 (Fig. 3e). Clearly, the theoretical prediction is reproduced very well for both nanochannels, despite their different cross-sectional dimensions and despite the different shapes of the imaged proteins and DNA molecules.

In a similar fashion, we derive the mean values of $D$ ($\overline{D}$) and the resolution in $D$ and compare the dependency of $\overline{D}$ of the protein monomers on their hydrodynamic radii taken from the literature[32] ($R_s$ = 8.6 nm for thyroglobulin, $R_s$ = 6.1 nm for ferritin, $R_s$ = 4.6 nm for ADH and $R_s$ = 3.5 nm for bovine serum albumin (BSA)) with the confinement-corrected Stokes–Einstein equation (Eq. 2) (Fig. 3f). We find that the theoretically predicted values reproduce the experimentally measured $\overline{D}$ values well, which suggests that potential surface-related effects are negligible.

Further analysis of the data library also reveals molecular populations with higher MW and higher $R_s$ than expected for monomers.

Specifically, for thyroglobulin and BSA, we observe indications of a second population with approximately double the MW and approximately 1.3× higher $R_s$, (marked 'Dim' in Fig. 3a,b for thyroglobulin, in Fig. 3c for BSA) that according to theory of asymmetric particle diffusion[33] corresponds to dimers. For ferritin, we also find dimers and, interestingly, also multiple subpopulations characterized by MW values somewhere between the monomer and dimer populations that correspond to molecules with different numbers of coordinated iron (details in Section 11 in Supplementary Information).

We also highlight that $\overline{iOC}$ and $\overline{D}$ values determined by ML are in excellent agreement with SA (Fig. 3e,f), as well as their distributions (inset of Fig. 3c for BSA and Extended Data Fig. 4 for all biomolecules). This validates the use of ML and we thus use it exclusively from here forward since it is substantially more efficient in terms of computational time.

**Surface passivation by supported lipid bilayer.** So far, we had designed our experiments such that the biomolecules were predominantly negatively charged to minimize attractive interaction—and thus nonspecific binding—to the negatively charged nanochannel walls. Nevertheless, we were able to observe rare events of molecular binding and unbinding to and from the nanochannel, respectively (Extended Data Fig. 5). For the data presented above, we excluded these transient binding events from the analysis (see Section 6 in Supplementary Information). At the same time, we also note that the demonstrated observation of nonspecific binding events opens the door to using NSM for affinity-based single molecule detection by analyte-specific receptors immobilized on nanochannel walls[34].

To now demonstrate the active prevention of nonspecific binding, we applied a supported lipid bilayer (SLB) coating[35] on the nanochannel walls, which is formed by adsorption and subsequent rupturing of large unilamellar vesicles (LUVs) (Fig. 4a; details in 'Supported lipid bilayer' in Methods). The real-time NSM-response to the SLB formation inside a $225 \times 200$ nm² nanochannel (Channel V; Extended Data Fig. 3e) is shown in Fig. 4b. To demonstrate the surface-passivation effect of the SLB, we also coated an $82 \times 40$ nm² nanochannel (Channel VI; Extended Data Fig. 3f) and used it to successfully characterize the positively charged

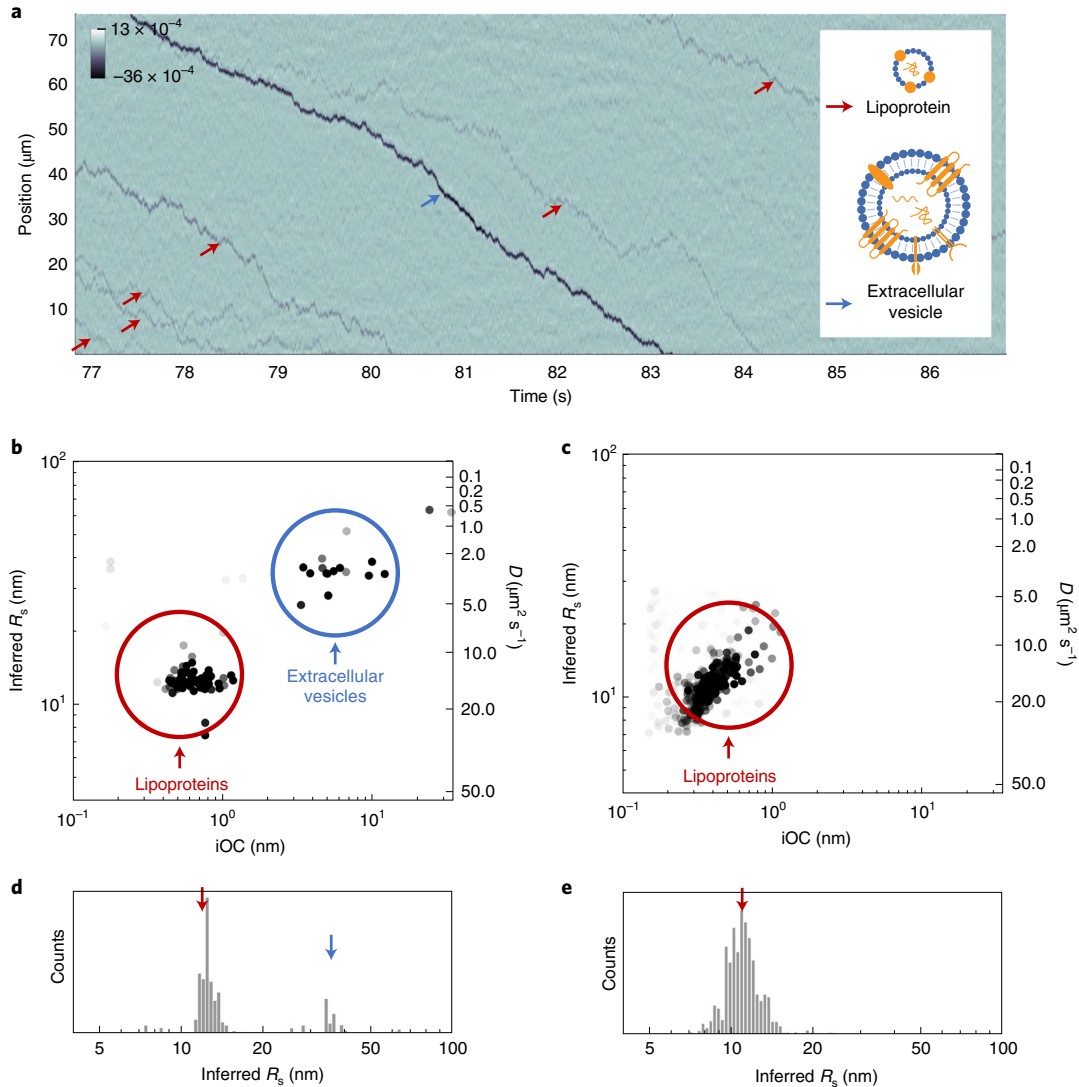

**Fig. 5 | Analysis of BNPs in conditioned cell culture medium containing serum. a**, Kymograph of multiple lipoprotein particles (red arrows) and one larger EV (blue arrow) moving through the nanochannel. Inset, schematics of an EV and a lipoprotein (depicted to scale). **b,c**, Scatter plots of iOC and $D$ translated into $R_s$. **d,e**, Histograms of $R_s$, analyzed using ML. Data correspond to the conditioned SH-SY5Y human cell medium (**b,d**) and to the control where the medium had not been in contact with the cells (**c,e**). All data were acquired in Channel V, which had been passivated by an SLB before measurement.

protein aldolase (Fig. 4c), which was impossible to analyze in an uncoated channel due to the strong electrostatic interaction with the negatively charged channel walls. The obtained values for aldolase are in good agreement with the nominal values (Fig. 4d,e) and thus corroborate the wide applicability of NSM, irrespective of analyte charge.

**Analysis of extracellular vesicles in cell culture medium.** EVs act as mediators of physiological intercellular communication, play key roles in the pathobiology of several diseases[36] and are promising diagnostic biomarkers[37]. Their functionality depends on both their composition and their size. However, these parameters are challenging to define precisely with existing methodologies, such as dynamic light scattering or nanoparticle tracking analysis (NTA), due to the substantial heterogeneity of EVs, in combination with their small sizes down to the tens of nanometers range[38], and their existence in complex biofluids.

Here, to apply NSM to complex biological sample analysis, we collected conditioned medium from human SH-SY5Y cells (details

in 'Conditioned cell culture medium' in Methods), containing a mixture of serum proteins and secreted EVs. To accommodate the size of EVs, we used a $225 \times 200\,nm^2$ nanochannel (Channel V), whose walls were passivated with an SLB to prevent nonspecific binding. To increase the throughput to up to ten particles per minute, we introduced a slow flow by applying a 0.25 Pa pressure drop along the nanochannel, resulting in individual particles being flushed through it, as revealed by the NSM signal (Fig. 5a).

Using this setup, we then collected a significant number of trajectories of different BNPs, from which we derive iOC and $D$ translated to $R_s$ using Eq. 2 (the applicability of the no-slip condition is discussed in Section 4 in Supplementary Information). This analysis reveals two distinct populations (Fig. 5b): one in the small iOC/$R_s$ regime (red circle) corresponding to most trajectories, and one in the large iOC/$R_s$ regime (blue circle) corresponding to a minority of trajectories. A control experiment with cell culture medium that had not been in contact with the cells reveals that the former population can be attributed to entities present in the serum supplement, probably lipoprotein particles (Fig. 5c), whereas the second

population indeed corresponds to EVs secreted by the cells. The identified $R_S$ of 10–13 nm and 20–70 nm (Fig. 5d,e) correspond to values reported for low-density lipoproteins[39] and EVs[38], respectively. In addition, the obtained size of EVs was validated by a comparative size distribution measurement using NTA (Supplementary Fig. 20; more details in 'NTA' in Methods). This clearly demonstrates the applicability of NSM to detect and, importantly, distinguish nonlabelled analytes of biological relevance in a complex sample mixture retrieved from cell culture. However, we also note that precise translation of iOC into MW is more complicated for BNPs than for single molecules due to a large variety of molecular constituents with different optical properties whose representation and spatial distribution might be different for each BNP (that is, different constant $a$ in Eq. 1). Therefore, we did not further explore the content of the EVs via the measured iOC.

## Discussion

We have introduced NSM to quantitatively analyze single biomolecules without the need for labeling or surface attachment, as they diffuse inside a nanofluidic channel. NSM provides spatio-temporal information about MW via optical contrast for biomolecules down to 66 kDa, and reveals $R_s$ and molecular conformation via molecular diffusivity $D$. Furthermore, ferritin monomers with different iron content could be identified, showing the ability of the method to distinguish molecular subpopulations within a sample. Moreover, SLB coating on the nanochannel walls prevented nonspecific binding, thereby enabling analysis of both positively and negatively charged proteins. These key NSM attributes challenge existing label-free single biomolecule detection techniques, which all require analyte binding to a surface for detection, and enable up to threefold improvement in terms of MW resolution compared with state-of-the-art methods. Furthermore, NSM complements single molecule fluorescence microscopy in cases where labeling is unwanted or even impossible. Thus, NSM will find application in studies of conformational changes, aggregation processes or reactions between individual biomolecules, and help unravel the dynamics of such processes and molecule-specific heterogeneities that might be of physiological relevance[40].

As a second aspect, we have analyzed conditioned cell culture medium collected from human SH-SY5Y cells, which contained a complex mixture of proteins and BNPs. We were able to accurately determine and, importantly, distinguish the size and corresponding distributions of lipoprotein particles and EVs in this setting. Looking forward, this advertises NSM for label-free single-cell studies in real time, such as for the analysis of intracellular content or secretomes, for which we predict it to be particularly useful due to minimized sample dilution in the nanofluidic system. Furthermore, rapid progress in detector technologies and ML-based data treatment promises to push the limits of NSM towards even smaller molecules and to enable up to two orders of magnitude increased throughput via parallel analysis of hundreds of nanochannels that fit in the field of view of a microscope[41]. Therefore, we predict NSM to find applications across a wide range of fields, including genomic DNA analysis[42], single particle counting[43] and single particle catalysis[44].

## Online content

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

## Methods

**Biomolecular solutions.** Alcohol dehydrogenase (ADH) from *Saccharomyces cerevisiae* (MW = 150 kDa; pI = 5.4) and aldolase from rabbit muscle (MW = 158 kDa; pI = 8.16) was purchased from Sigma-Aldrich. Ferritin (MW = 440 kDa; pI = 5.1–5.7) and thyroglobulin (MW = 669 kDa; pI = 4.7) were purchased from GE Healthcare. A concentration of 28 nM of the protein solution in PBS (pH 7.4, Sigma-Aldrich) was used. DNA fragments (MW = 650 Da × number of base pairs) and BSA (MW = 66 kDa, pI = 4.7) were purchased from ThermoFisher Scientific; a concentration of 28 nM of the DNA solution in 0.05× TBE (ThermoFisher Scientific) was used.

**Supported lipid bilayer.** 1-Palmitoyl-2-oleoyl-glycero-3-phosphocholine (POPC) was purchased from Avanti Polar Lipids. Marina blue 1,2-dihexadecanoyl-sn-glycero-3-phosphoethanolamine (Marina Blue DHPE) was purchased from ThermoFisher Scientific. PBS (tablets) were purchased from Sigma-Aldrich. PBS buffers were filtered with Millipore filters (0.1 μm) from Merck and water was deionized and filtered using a Milli-Q system from Merck.

LUVs were prepared by mixing lipids (POPC/Marina Blue (99/1 mol%)) in round-bottom flasks followed by drying of the lipid films overnight under vacuum. The lipid films were rehydrated and mixed with PBS to a total lipid concentration of 1 mg ml⁻¹. Afterwards, rehydrated lipids were disrupted by five freeze-thaw cycles using liquid nitrogen and a 37 °C heat block except for the last cycle, which was thawed at room temperature. Finally, LUVs were obtained by extrusion using a mini-extruder (Avanti Polar Lipids), with two polycarbonate filters of 100 nm pore diameter (21 passages). To form a SLB in nanochannels of size 225 × 200 nm² (Channel V), we let the LUV solution flow through the nanochannels for 30 min by applying the pressure drop of 2 bar. For nanochannel of size 80 × 40 nm² (Channel V), the process was prolonged to 12 h. In addition, to prevent eventual clogging, the direction of the flow was switched periodically with a period of 10 s using microfluidic pressure controller MFCS-EX (Fluigent).

**Conditioned cell culture medium.** Fetal bovine serum (FBS), Ham's F-12 nutrient mixture (F12), DMEM GlutaMAX, nonessential amino acids (NEAA) and trypsin-EDTA were purchased from ThermoFisher Scientific. Amicon Ultra-15 centrifugal filter unit 100 kDa 15 ml and PBS (tablets) were purchased from Sigma-Aldrich. PBS buffers were filtered with Millipore filters (0.1 μm) from Merck and water was deionized and filtered using a Milli-Q system from Merck.

SH-SY5Y human neuroblastoma cells (1.2 × 10⁶) were seeded in T25 flasks (ThermoFisher Scientific) in 5 ml cell culture medium containing a 1:1 mix of MEM and F12 supplemented with 10% FBS and 1% NEAA and cultured for 48 h at 37 °C with 5% CO₂. The conditioned medium was then harvested, filtrated through a 0.22 μm filter (VWR) and centrifuged for 20 min at 4,000 rpm at 4 °C in an Eppendorf Centrifuge 5430/5430 R with Rotor F-35–6–30 to remove large cell debris. The supernatant was then loaded onto Amicon Ultra-15 100 kDa filters (pre-equilibrated with 5 ml filtered PBS for 10 min at 5,680 rpm at 4 °C) to concentrate EVs and protein particles. The Amicon tubes were centrifuged for 2 h at 6,350 rpm, 4 °C. Afterwards, flowthrough was discarded and Amicon tubes were loaded with 5 ml filtered PBS and centrifuged for 2 h at 6,350 rpm, 4 °C. Finally, the concentrated conditioned cell culture medium was collected from the filter. For the control experiment, 5 ml cell culture medium that had not been in contact with the cells was concentrated on Amicon filters using the same protocol. For NSM measurements the samples were diluted 130× in PBS to reach a sufficiently low particle concentration.

**Nanofluidic chips: design.** The nanofluidic chips used in our experiments (Extended Data Fig. 2) contain a series of nanochannels with tailored cross-sectional dimensions and 30 μm length (Channels I–IV and VI) and 200 μm length (Channel V), which are connected to macroscopic in- and outlets through two microchannels with cross-section dimensions of 50 × 1.5 μm². The transport of the 30 μl liquid sample from the inlets to the nanochannels is controlled by pressurizing the inlets to 2 bar. During the measurements in Channels I–IV and VI, the applied pressure is turned off to solely rely on diffusion for molecular motion through the nanochannels during imaging. The dataset corresponding to the ADH molecule in Channel I was collected from three different nanochannels of the same geometry and the same nanofluidic chip. No statistically relevant differences between the results from different nanochannels were found. The other data sets corresponding to a combination of a specific biomolecule and a specific nanochannel were collected from the same volume of biomolecular solutions and the same nanochannel.

**Nanofluidic chips: nanofabrication.** Fabrication of the nanofluidic chips was carried out in cleanroom facilities of Federal Standard 209 E Class 10–100, using electron-beam lithography (JBX-9300FS/JEOL Ltd), Ion-beam etching (Ionfab 300 Plus/Oxford Plasma Technology), photolithography (MA6/Suss MicroTec), reactive-ion etching (Plasmalab 100 ICP180/Oxford Plasma Technology and STS ICP and PlasmaTherm/Advanced Vacuum), electron-beam evaporation (PVD 225/Lesker and HVC600/AVAC), magnetron sputtering (MS150/FHR), deep reactive-ion etching (STS ICP/STS) and wet oxidation (wet oxidation/centrotherm), fusion bonding (AWF 12/65/Lenton), scanning electron microscopy (Supra 55VP/Zeiss) and dicing (DAD3350/Disco). In particular, the fabrication comprised the following processing steps of a 4-inch silicon (p-type) wafer of 1 mm thickness:

Thermal oxidation: (1) cleaning for 10 min at 80 °C in 1:1:5 H₂O₂:NH₃OH:H₂O (SC-1), rinsing in water, HF-dip for 30 s, cleaning for 10 min at 80 °C in 1:1:5 H₂O₂:HCl:H₂O (SC-2), rinse in water and drying under N₂-stream. (2) Wet oxidation in water atmosphere for 660 min at 1,050 °C (2,000 nm thermal oxide).

Fabrication of alignment marks: (1) spin coating HMDS adhesion promoter (MicroChem) at 3,000 rpm for 30 s and soft baking on a hotplate (HP) at 115 °C for 120 s. Spin coating S1813 (Shipley) at 3,000 rpm for 30 s and soft baking (HP) at 110 °C for 1 min. (2) Expose alignment marks for 12 s in contact aligner at 6 mW cm⁻² intensity. (3) Development in MF-319 (Microposit) for 60 s, rinsing in water and drying under N₂-stream. (4) Reactive-ion etching (RIE, PlasmaTherm) for 15 s at 250 mTorr chamber pressure, 50 W RF-power and 80 sccm O₂-flow (descum). RIE for 45 min at 100 mTorr chamber pressure, 100 W RF-power and 40 sccm CF₄-flow (900 nm etch depth in thermal oxide). (5) Removal of resist in 50 ml H₂O₂ + 100 ml H₂SO₄ at 130 °C for 10 min, rinsing in water and drying under N₂-stream.

Fabrication of nanochannels: (1) Electron-beam evaporation of 12 nm Cr (hard mask). (2) Spin coating 950k-PMMA A4 (MicroChem Corporation) at 6,000 rpm for 60 s and soft baking (HP) at 180 °C for 5 min. (3) Electron-beam exposure at 1 nA with a shot pitch of 2 nm and 600 μC cm⁻² exposure dose. (4) Development in IPA 4: 1 H₂O for 45 s at 6 °C and drying under N2-streaI. (5) Ion-beam etching at 10 mA beam current, 300 V beam voltage, 300 V acceleration voltage and 200 mA neutralizer current for 16 min. (6) RIE (Plasmalab) for 150 s at 2 mTorr chamber pressure, 40 W RF-power and 50 sccm NF3-flow. (7) Removal of Cr-mask in 50 ml H₂O₂ + 100 ml H₂SO₄ at 130 °C for 10 min, rinsing in water, wet-etching in standard Cr-wet etch and drying under N₂-stream.

Fabrication of microchannels: (1) spin coating HMDS at 3,000 rpm for 30 s and soft baking (HP) at 115 °C for 2 min. Spin coating S1813 (Shipley) at 3,000 rpm for 30 s and soft baking (HP) at 110 °C for 1 min. (2) Expose microchannels for 8 s in contact aligner at 6 mW cm⁻² intensity. (3) Development in MF-319 (Microposit) for 60 s, rinsing in water and drying under N₂-stream. (4) RIE for 15 s at 60 mTorr chamber pressure, 60 W RF-power and 60 sccm O₂-flow (descum). RIE for 20 min at 30 mTorr chamber pressure, 275 W RF-power, 50 sccm Ar-flow and 50 sccm CHF₃-flow (1,500 nm etch depth in thermal oxide). (5) Removal of resist in 50 ml H₂O₂ + 100 ml H₂SO₄ at 130 °C for 10 min, rinsing in water and drying under N₂-stream.

Fabrication of inlets (from backside): (1) magnetron sputtering of 500 nm Al (hard mask). (2) Spin coating S1813 at 3,000 rpm for 30 s and soft baking (HP) at 110 °C for 1 min. (3) Expose inlets for 12 s in contact aligner at 6 mW cm⁻² intensity. (4) Development in MF-319 for 60 s, rinsing in water and drying under N₂-streaI. (5) Aluminum wet etch (4:4:1:1 H₃PO₄: CH₃COOH: HNO₃: H₂O) for 10 min to clear the hard mask at inlet positions. (6) RIE for 30 min at 30 mTorr chamber pressure, 275 W RF-power, 50 sccm Ar-flow, 50 sccm CHF3-flow. (7) Deep RIE for 1,500 cycles of 12 s at 5 mTorr chamber pressure, 600 W RF-power, 10 W platen power, 130 sccm SF6-flow (Si-etch) and 7 s at 5 mTorr chamber pressure, 600 W RF-power, 10 W platen power and 85 sccm C4F8-flow (passivation). (8) Removal of Al-hard mask in 50 ml H₂O₂ + 100 ml H₂SO₄ at 130 °C for 10 min, rinsing in water and drying under N₂-stream.

Fusion bonding: (1) cleaning of the substrate together with a lid (175 μm thick 4-inch pyrex, UniversityWafers) in 5:1:1 H₂O:H₂O₂:NH₃OH (SC-1) for 10 min at 80 °C. (2) Prebonding the lid to the substrate by bringing surfaces together and applying pressure manually. (c) Fusion bonding of the lid to the substrate for 5 h in N₂ atmosphere at 550 °C (5 °C min⁻¹ ramp rate).

Dicing of bonded wafers: cutting nanofluidic chips from the bonded wafer using a resin-bonded diamond blade of 250 μm thickness (Dicing Blade Technology) at 25 krpm and 2 mm s⁻¹ feed rate.

**Nanofluidic chips: scanning electron microscopy images.** The nanofluidic chips were prediced from the back side to a depth of 400 μm at the position of the nanochannel arrays, using a resin-bonded diamond blade of 100 μm thickness (Dicing Blade Technology) at 35 krpm and 5 mm s⁻¹ feed rate. The chips were then cleaved manually and 3 nm C was deposited on the cleaved surfaces with electron-beam evaporation (HVC600) for discharging during scanning electron microscopy (SEM) imaging. The nanochannels were imaged in cross-section with SEM at 15 kV beam voltage and a working distance of 2.5 mm using an in-lens detector.

To measure the cross-sectional area from the SEM images, the edges of the nanochannels were estimated manually from the positions of the pixels with the most rapid change of the intensity (dashed line in Extended Data Fig. 3) and drawn on top of the SEM image using a computer graphics tool (Affinity Designer). A secondary image containing only the derived shape of a nanochannel with dark pixels inside the area of the nanochannel and white pixels outside the area of the nanochannel was then created. The number of black pixels was then calculated using Matlab and translated into micrometers squared accordingly. The area derived ($A$) was translated into a format expressing the width × length that correspond to the dimensions of an equal-area rectangle. The measured areas of Channels I–VI correspond to $A_I = 100 \times 27$ nm², $A_{II} = 110 \times 72$ nm², $A_{III} = 100 \times 15$ nm², $A_{IV} = 145 \times 27$ nm², $A_V = 225 \times 200$ nm² and $A_{VI} = 82 \times 40$ nm², respectively.

**Experimental setup.** The optical setup was based on the versatile microscope platform RM21 (Mad City Labs) (Extended Data Fig. 1). A beam of collimated polychromatic light with wavelength range 400–2,350 nm generated by a supercontinuum laser (NKT Photonics, SuperK EXTREME EXB-6) was spectrally filtered by the tunable wavelength filter (NKT Photonics, SuperK VARIA) to a polychromatic beam with wavelength range 450–750 nm. The resulting beam with a total power of 250 mW was focused at the back focal plane of a microscope objective lens (numerical aperture (NA) = 1.49, Nikon), and directed to illuminate the fluidic chip under angle using a micromirror positioned at the back aperture of the objective lens. The effective NA of the objective was then lowered to 1.27 as the back aperture of the objective was partially blocked—decreased from a diameter of 23.8 mm to 16.7 mm. The reflected light was spatially filtered using a second micromirror and the scattered light was imaged via a tube lens by a CMOS camera (Andor, Zyla). The resulting magnification was 220×. In the case of the experiments with 30 μm long nanochannels (Channels I–IV and VI), a beam of 1 mm diameter was used to illuminate an area of the fluidic chip of 10 μm diameter. An area of 30 × 600 pixels of the camera was then acquired at the frame rate of about 5,000 frames per second. In the case of the experiments with 200 μm long nanochannels (Channel V), the beam was expanded fourfold to illuminate an area of the fluidic chip of 40 μm diameter. An area of 30 × 2,160 pixels of the camera was then acquired at the same frame rate.

**Nanoparticle tracking analysis.** We performed NTA using a Malvern NanoSight LM10 instrument equipped with a 488 nm laser and operating in scattering mode and under a flow rate of 100 (B10 ml min⁻¹) obtained with a Nanosight syringe pump module and with the cameral level set to 15. The sample was diluted 1,000× in PBS and analyzed in a set of five videos of 60 s each. The videos were analyzed with the built-in NTA 3.2 software using a detection threshold of five to be able to determine optimized size distributions and concentrations. The buffer viscosity was considered as that of water at 21 °C. Concentration values (particles per milliliter) were extracted and plotted.

**Reporting Summary.** Further information on research design is available in the Nature Research Reporting Summary linked to this article.

## Data availability
All sample data are packaged with the code and can be found at gitlab.com/langhammerlab/NSM-SA. Source data are provided with this paper.

## Code availability
The codes for device control and data acquisition can be found at gitlab.com/langhammerlab/NSM-DeviceControl. The code using SA described in Sections 5–6 in Supplementary Information can be found at gitlab.com/langhammerlab/NSM-SA. The code using ML described in Section 7 in Supplementary Information can be found at gitlab.com/langhammerlab/NSM-ML.

## Acknowledgements
The authors acknowledge financial support from the Swedish Foundation for Strategic Research project FFL15-0087 (C.L.), the Knut and Alice Wallenberg Foundation project 2015.0055 (C.L.) and the European Research Council for ERC Consolidator grant project 866238 (F.W.) Part of this research has been executed at the Chalmers Nanofabrication Laboratory MC2 and under the umbrella of the Chalmers Excellence Initiative Nano.

## Author contributions
B.S., J.F. and C.L. conceived the project. B.S. developed the NSM analytical theory and provided FDTD simulations. B.S. built the optical setup based on the MadCityLab microscopy platform. J.T. wrote the device control software. J.F. and D.A. produced the nanofluidic chips. B.S., J.T. and D.A. performed the NSM imaging. B.S. designed and implemented SA. H.K.M. and G.S. designed and implemented ML analysis with guidance from D.M. and G.V. B.S., H.K.M. and G.S. analyzed the data. B.S. and H.S.-J. designed the experiments with biomolecular solutions and prepared the samples. Q.L. and D.v.L. prepared the cell culture media and LUV solutions with guidance from E.K.E. D.L. provided the NTA measurements. F.W., D.M., E.K.E., M.K., G.V. and C.L. contributed to data interpretation and provided valuable feedback. B.S. and C.L. wrote the paper with input from all authors. C.L. provided the funding for the project.

## Funding

## Competing interests
B.S., J.F. and C.L. are founders of Envue Technologies AB, which owns intellectual property related to the described research. J.F. and C.L. are members of the board of Envue Technologies AB. The remaining authors declare no competing interests.

## Additional information
**Extended data** is available for this paper at https://doi.org/10.1038/s41592-022-01491-6.

**Correspondence and requests for materials** should be addressed to Barbora Špačková or Christoph Langhammer.

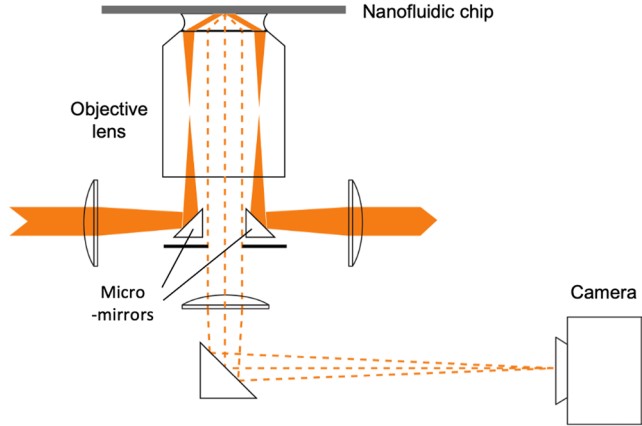

**Extended Data Fig. 1 |** Schematic of the experimental setup.

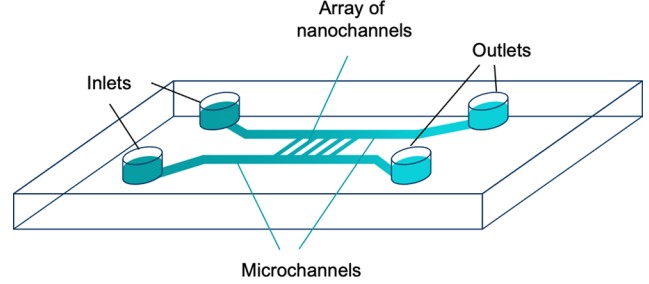

**Extended Data Fig. 2 |** Schematic of the nanofluidic chip.

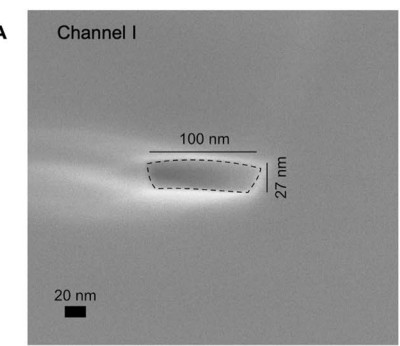

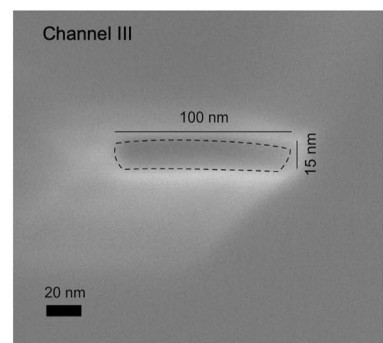

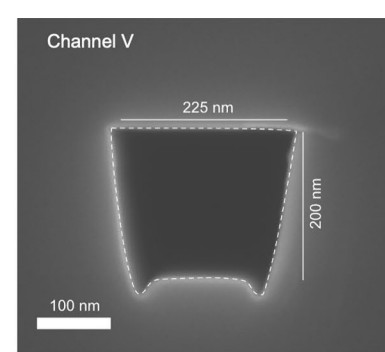

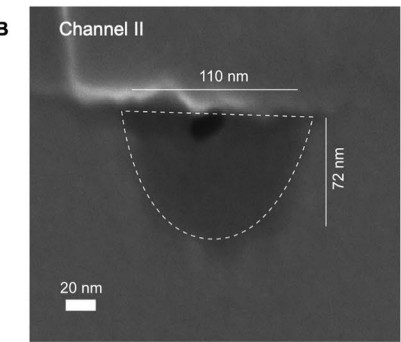

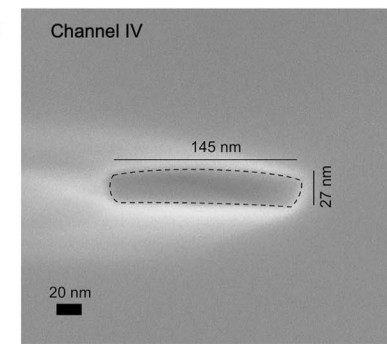

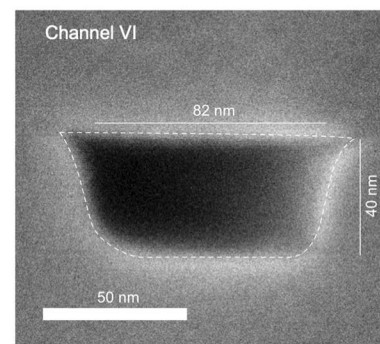

**Extended Data Fig. 3 | SEM images of nanochannel cross sections of (A-F) Channel I – VI used in different experiments.** The domain from which the area of the cross section was calculated is highlighted by the dashed line. The width and length correspond to the dimensions of a rectangle with area equal to the determined cross-sectional area of the nanochannel.

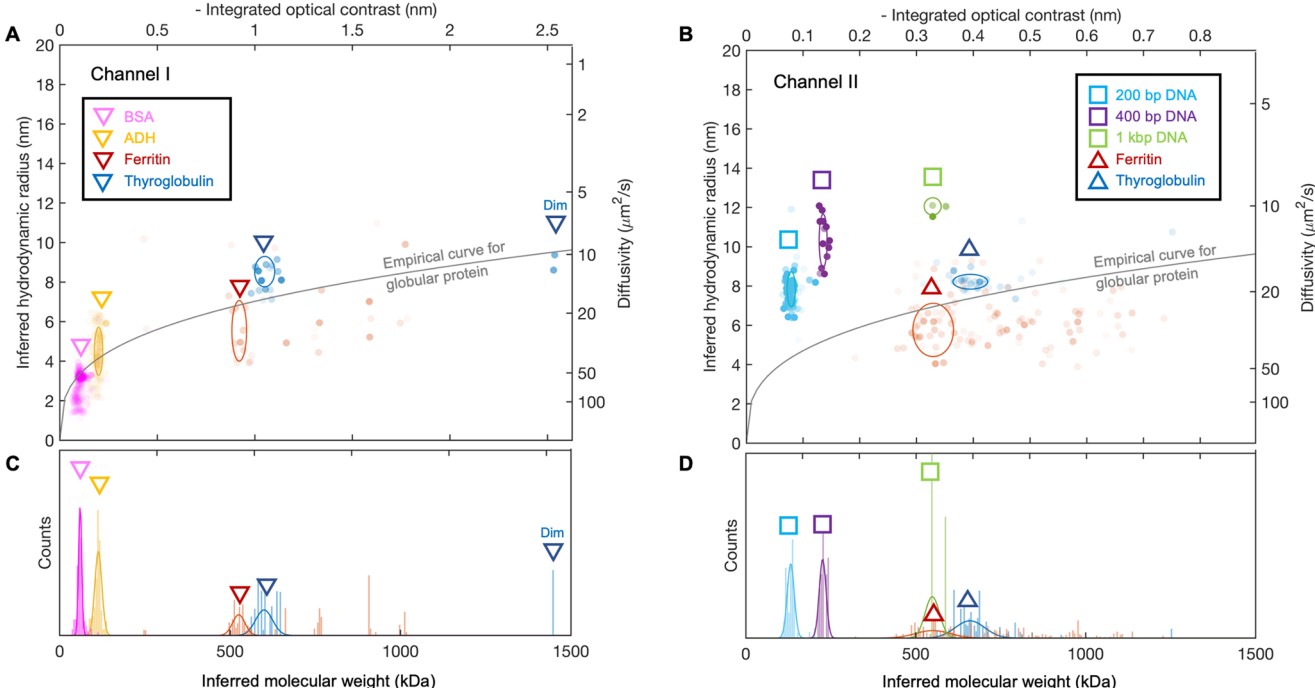

**Extended Data Fig. 4 | Machine-learning analysis results. (A, B)** Scatter plots of iOC translated into MW using Eq. 1 in the main text, and D translated into $R_s$ using Eq. 2 in the main text for individual biomolecules of different types, measured in **(A)** Channel I and **(B)** Channel II using ML. Each dot is extracted from a single biomolecule trajectory. Color intensity scales linearly with frame number of the trajectory (N). The highest intensity corresponds to N = 2130 frames. The gray line corresponds to an empirical relation between MW and D for globular protein[30] **(C, D)** iOC histograms of the biomolecules in **(A, B)** translated into MW using Eq. 1 in the main text.

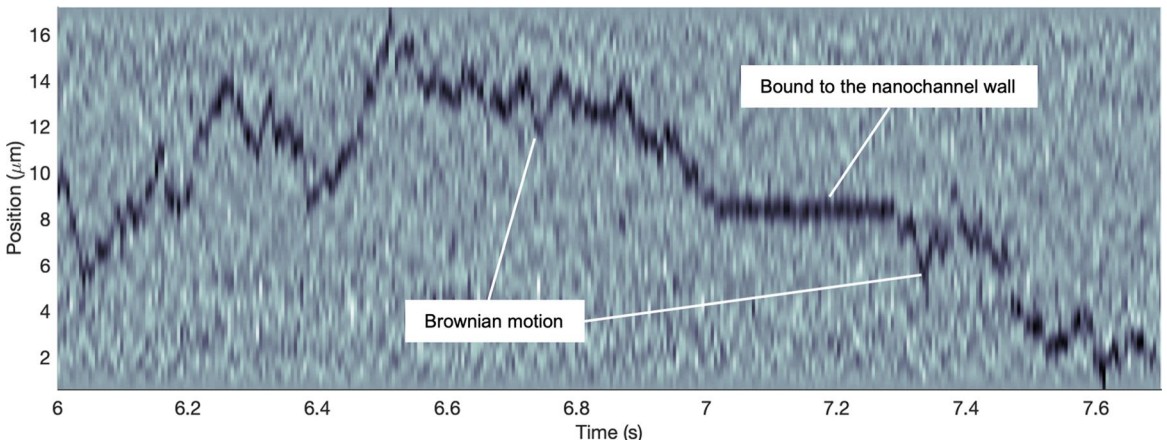

**Extended Data Fig. 5 | Ferritin–nanochannel wall interaction.** Kymograph of a single ferritin molecule inside Channel II. For each time frame, the dark spot in the image corresponds to the position of the ferritin that resides inside the field of the view. The position of the biomolecule changes stochastically in the left and right part of the image, whereas in the central part, it remains fixed for about 0.3 s. These two different behaviors correspond to two different states of a biomolecule – freely diffusing and bound to the nanochannel wall, respectively. The trajectory of the biomolecule was found using a particle tracking algorithm and from the statistics of the movement, the part of the trajectory corresponding to the bound state was identified and excluded from the further analysis (more details in SI section "Particle tracking").

# nature research

# Reporting Summary

Nature Research wishes to improve the reproducibility of the work that we publish. This form provides structure for consistency and transparency in reporting. For further information on Nature Research policies, see our Editorial Policies and the Editorial Policy Checklist.

## Statistics

For all statistical analyses, confirm that the following items are present in the figure legend, table legend, main text, or Methods section.

| n/a | Confirmed | |
|---|---|---|
| ☐ | ☒ | The exact sample size (*n*) for each experimental group/condition, given as a discrete number and unit of measurement |
| ☐ | ☒ | A statement on whether measurements were taken from distinct samples or whether the same sample was measured repeatedly |
| ☒ | ☐ | The statistical test(s) used AND whether they are one- or two-sided<br>*Only common tests should be described solely by name; describe more complex techniques in the Methods section.* |
| ☒ | ☐ | A description of all covariates tested |
| ☒ | ☐ | A description of any assumptions or corrections, such as tests of normality and adjustment for multiple comparisons |
| ☐ | ☒ | A full description of the statistical parameters including central tendency (e.g. means) or other basic estimates (e.g. regression coefficient) AND variation (e.g. standard deviation) or associated estimates of uncertainty (e.g. confidence intervals) |
| ☒ | ☐ | For null hypothesis testing, the test statistic (e.g. *F*, *t*, *r*) with confidence intervals, effect sizes, degrees of freedom and *P* value noted<br>*Give P values as exact values whenever suitable.* |
| ☒ | ☐ | For Bayesian analysis, information on the choice of priors and Markov chain Monte Carlo settings |
| ☒ | ☐ | For hierarchical and complex designs, identification of the appropriate level for tests and full reporting of outcomes |
| ☒ | ☐ | Estimates of effect sizes (e.g. Cohen's *d*, Pearson's *r*), indicating how they were calculated |

*Our web collection on statistics for biologists contains articles on many of the points above.*

## Software and code

Policy information about availability of computer code

| Data collection | The data were collected using a custom code NSM-DeviseControl v1.0 that can be found at gitlab.com/langhammerlab/NSM-DeviceControl. |
|---|---|
| Data analysis | The data were analyzed by a custom code NSM-SA v 1.0 utilizing SA described in SI sections 5-6 and a custom code NSM-ML v1.0 utilizing ML described in SI section 7. The codes can be found at gitlab.com/langhammerlab/NSM-SA and gitlab.com/langhammerlab/NSM-ML. |

For manuscripts utilizing custom algorithms or software that are central to the research but not yet described in published literature, software must be made available to editors and reviewers. We strongly encourage code deposition in a community repository (e.g. GitHub). See the Nature Research guidelines for submitting code & software for further information.

## Data

Policy information about availability of data

All manuscripts must include a data availability statement. This statement should provide the following information, where applicable:
- Accession codes, unique identifiers, or web links for publicly available datasets
- A list of figures that have associated raw data
- A description of any restrictions on data availability

A sample data are packaged with the code and can be found at gitlab.com/langhammerlab/NSM-SA. Source data are provided with this paper.

# Field-specific reporting

Please select the one below that is the best fit for your research. If you are not sure, read the appropriate sections before making your selection.

☒ Life sciences ☐ Behavioural & social sciences ☐ Ecological, evolutionary & environmental sciences

For a reference copy of the document with all sections, see nature.com/documents/nr-reporting-summary-flat.pdf

# Life sciences study design

All studies must disclose on these points even when the disclosure is negative.

| | |
|---|---|
| Sample size | This is not relevant to us because we draw no "life science" conclusions based on the results. We only report the information of the measured sample without any general conclusions about samples taken from the natural world. |
| Data exclusions | No data were excluded. |
| Replication | To verify the reproducibility, the data corresponding to the ADH molecule were collected using three different nanochannels with the same geometry. No statistically relevant differences between the results from different nanochannels were found. |
| Randomization | No randomization was done because we have not performed any "treatment" or modification of samples. The samples are taken from specific sources outlined in the main text. |
| Blinding | Same as randomization. |

# Reporting for specific materials, systems and methods

We require information from authors about some types of materials, experimental systems and methods used in many studies. Here, indicate whether each material, system or method listed is relevant to your study. If you are not sure if a list item applies to your research, read the appropriate section before selecting a response.

### Materials & experimental systems

| n/a | Involved in the study |
|---|---|
| ☒ | ☐ Antibodies |
| ☐ | ☒ Eukaryotic cell lines |
| ☒ | ☐ Palaeontology and archaeology |
| ☒ | ☐ Animals and other organisms |
| ☒ | ☐ Human research participants |
| ☒ | ☐ Clinical data |
| ☒ | ☐ Dual use research of concern |

### Methods

| n/a | Involved in the study |
|---|---|
| ☒ | ☐ ChIP-seq |
| ☒ | ☐ Flow cytometry |
| ☒ | ☐ MRI-based neuroimaging |

## Eukaryotic cell lines

Policy information about cell lines

| | |
|---|---|
| Cell line source(s) | SH-SY5Y human neuroblastoma acquired from ECACC. |
| Authentication | The SH-SY5Y cell line was authenticated by Eurofin Genomics by STR profiling according to ANSI/ATCC standard ASN-002. |
| Mycoplasma contamination | The SH-SY5Y cells were regularly tested for mycoplasma contamination by standardised qPCR test by Eurofins Genomics under ISO17025 accreditation. The test results were always negative showing that the cell cultures used are mycoplasma free. |
| Commonly misidentified lines (See ICLAC register) | None; our cell line is not in the ICLAC register. |

