## [Peer Review File · Nature Methods]

Peer Review Information

Manuscript Title: "Label-Free Nanofluidic Scattering Microscopy of Size and Mass of Single Diffusing Molecules and Nanoparticles"

Corresponding author name(s): Christoph Langhammer, Barbora Špačková

Reviewer Comments & Decisions:

Decision Letter, initial version:

Dear Professor Langhammer,

Your Article entitled "Nanofluidic Scattering Microscopy for Label-free Weight and Size Screening of Single Diffusing Biomolecules" has now been seen by three reviewers, whose comments are attached. While they find your work of some potential interest, they have raised concerns which in our view are sufficiently important that they preclude publication of the work in Nature Methods.

We will consider looking at a revised manuscript only if further experimental data allow you to address all the major criticisms of the reviewers (unless, of course, something similar has by then been accepted at Nature Methods or appeared elsewhere). This includes submission or publication of a portion of this work somewhere else.

Our biggest concern is that the data collected will not be biologically meaningful because of the confines of the channel. We are also concerned that the method seems only applicable to negatively charged proteins. If you can solidly rebut these concerns with new experimental evidence while addressing the other technical concerns, we would be willing to consider whether to send your paper back to the referees for further consideration.

If you are interested in revising this manuscript for submission to Nature Methods in the future, please ****contact me to discuss your appeal**** before making any revisions. Otherwise, we hope that you find the reviewers' comments helpful when preparing your paper for submission elsewhere.

Sincerely,

Rita

Rita Strack, Ph.D.
Senior Editor
Nature Methods

Reviewers' Comments:

Reviewer #1:

Remarks to the Author:

This manuscript presents nanofluidic scattering microscopy (NSM), a label-free approach for solution phase single molecule detection in a confined nanochannel, providing the information about the molecular mass, size, and diffusivity, based on interference scattering imaging. This is an interesting work, as the use of nano channel is a clever way for stable imaging and tracking single molecules in solution phase, which is distinct from most other label-free optical single molecule detection methods that can only measure molecules attached to a surface. The capability to measure the diffusion of the single molecule in the solution could be used to explore the molecular behavior in real biofluid (Nat Methods 17, 524–530 (2020)). The manuscript is well written in general. However, this work is lack of solid experimental demonstration for the claimed applications and clarification on its limitations. Thus, the manuscript is not ready for publication in current form. I have the following questions and suggestions:

1. What is the throughput of the detection? Compared to the wide field imaging approaches (Science 27 Apr 2018: Vol. 360, Issue 6387, pp. 423-427; Nat Methods 17, 1010–1017 (2020)), NSM relies on the confinement of nano channels. Trapping and measure one molecule a time in a nanochannel seems a low throughput. This could be a major limitation of the method. For some of the claimed potential applications, throughput is critical. For example, study of molecular interactions and the heterogeneities of molecular behavior and functions need adequate throughput to get statistically valid result. Also, low-throughput solution phase single molecule binding has been realized with optical tweezer (Analyst, 2015, 140, 4760-4778; Nano Lett. 2014, 14, 10, 5787–5791; ACS Photonics 2014, 1, 5, 389–393).
2. The behavior of the molecule in a nanochannel is different than in a bulk solution phase due to the surface effect (ACS Omega 2019, 4, 16, 17016–17030; J. Phys. Chem. B 2006, 110, 22, 10910–10918). A discussion should be given on how this difference will impact the anticipated applications of the method. For example, how much effect does the interaction between the channel wall and molecule on the measured diffusion coefficient?
3. Single molecule detection easily measures the false signals due to its high sensitivity, so a cross validation method should be provided to confirm the result. Such as the single molecule TIRF used by iSCAT (Nat Commun 5, 4495 (2014)), and antibody recognition used by plasmonic scattering imaging (Nat Methods 17, 1010–1017 (2020)).
4. Single molecules usually have high diffusion rate (Nat Methods 17, 524–530 (2020)). How to ensure the tracked molecule is the same molecule over time in the channel?

5. Is shot noise the dominant noise in the system as predicted by the calculation in the experimental setup section? A noise analysis similar to Fig. 2e in Nat Commun 5, 4495 (2014)) will answer this question.
6. The diffusion analysis relies on the localization, and the localization error is related to both signal to noise ratio and objective NA. Thus, a clear discussion about signal to noise ratio is needed. In addition, according to Fig. S1, the back aperture of objective is blocked partially. This will affect the effective NA (ACS Photonics 2017, 4, 2, 211–216; PNAS September 14, 2010 107 (37) 16028-16032). What is the effective objective NA in this optical configuration?
7. Interferometric image contrast for one molecule is determined by the phase. Different distance between the object and interference signal will lead to varying image contrast (Nat Methods 17, 1010–1017 (2020)). As shown in the FDTD simulation results (Figure S11 C and D), the scattering cross section of the biomolecules depends on the location of the molecule in the channel. Are the measured biomolecule signal intensities also showing similar pattern of location dependency? Is this location dependent intensity corrected for the determination of the molecular weight?
8. For the claimed NSM mass resolution in page 11, can a direct experimental proof be provided similar to iSCAT, by measure molecules with known added weights by modifying different numbers of biotins? (Science 27 Apr 2018: Vol. 360, Issue 6387, pp. 423-427)
9. Could the input and output for each step in the machine learning method (in Methods or in FigS7-9) be stated with determined size and 1D/2D profile? Can some intermediate results be given to help the reader understand the workflow?
10. If YOLO provides positions and intensities as its output, is FCNN part redundant?
11. Experimental demonstrations are missing for the claimed applications in the conclusions. These applications are hot topics, but they are non-trivial. For real biofluid detection, good surface chemistry is needed to discriminate the objects from impurities (ACS Sens. 2021, 6, 3, 1357–1366). For deep learning study, both the dataset and the theoretical model should be established well (Nat Methods 18, 194–202 (2021)). Can the authors pick up one of their claims to perform an example experiment? This is important to show the advantage of NSM.
12. Page 19, line 1, Ns is not defined.

Reviewer #2:

Remarks to the Author:

The manuscript presents an interferometric scattering imaging modality (Nanofluidic Scattering Microscopy, NSM) in which nanochannels are used to (1) confine the molecules of interest and (2) provide reference light field for interference. In comparison with conventional iSCAT techniques, the authors argue that NSM the major benefits of NSM are: (1) no need for surface attachment, and (2) prolonged sampling time. The authors claim that with these benefits, NSM can achieve 3-fold improvement on molecular weight resolution (30 kDa) comparing with the state-of-the-art iSCAT techniques.

However, based on the theory and data shown, I am not at all convinced that the authors' claims can be supported. To begin with, the authors claim that for iSCAT, "as key point, these three label-free optical single-molecule detection methods require the investigated species to bind to a surface to be 'visible'" (line 52-54), which is not true: In 2019, Taylor RW et al demonstrated via iSCAT the visualization of microsecond nanoscopic protein motion on a live cell membrane with high precision localization, in which the focal plane was placed "at a distance of several micrometres above the glass interface" (Nature Photonics 13.7 (2019): 480-487). In the manuscript, the authors confine the molecules of interest using nanochannels of different cross sections (Channel I: 100 nm × 27 nm, Channel II: 100 nm × 72 nm, Channel III: 100 nm × 15 nm, Channel IV: 145 nm × 27 nm, Fig. S3). The largest channel height is only 72 nm whereas the smallest channel height can be as small as 15 nm. In comparison, Taylor RW et al demonstrated that iSCAT can achieve an axial localization precision of 4-6 nm for particles within ~100 nm axial displacement via PSF fitting. In other words, the NSM technique actually need to bring the molecules closer to the surface than conventional iSCAT techniques.

There are additional limitations and concerns by utilizing the scattered light from nanochannels as the reference beam:

1. Clogging of molecules within the channel: the authors selectively choose negatively charged molecules in the study to prevent binding events on the channel walls. This greatly limits the practicality of NSM since in most study the molecules are not all negatively charged.
2. Difficulty in adjusting the optical contrast (iOC in the study) in interferometric scattering modality: $iOC = n \cdot \alpha_m / A$. To adjust iOC for small molecules, the only approach in the NSM method is to adjust A, which is the cross-sectional area of the nanochannel in use. In other words, the users would have to redesign and fabricate a customized nanochannel which involves e-beam lithography and thermal oxidation. In iSCAT, this can be simply achieved by swapping the gold dot attenuator of different extinction ratio.
3. Limited applications. NSM cannot be used for interferometric scattering imaging in 2D or 3D (for example, cell surface imaging), either can NSM be used for freely diffusive molecule tracking (due to the hindrance effect from the restricted volume of the nanochannel) which greatly limits the scope of the NSM technique as well the throughput (since the molecules are required to flow through a nanochannel rather than freely suspended in solution).
4. Additional costs for each imaging experiment by the nanochannel chips.

Additionally, the authors' claim on "3-fold improvement on molecular weight resolution" was based on a publication in 2019 (Science 360.6387 (2018): 423-427). However, for BSA the MW resolution is approximately the same. In addition, the current state-of-the-art iSCAT technique can achieve the same if not smaller resolution, for example, recently tubulin monomers of ~ 50 kDa was detected with a bandwidth of 5 kDa (Journal of molecular biology 432.23 (2020): 6168-6172.). BSA is of similar MW in the manuscript, yet the resolution reported using NSM is 20 kDa, which is much larger than the current state-of-the-art method. To claim a 3-fold improvement on molecular weight resolution, the authors need to compare with more than just one sample. The authors claim that the improvement was attributed to the prolonged sampling time. Yet I am suspecting that, since NSM uses polychromatic light,

the speckle patterns commonly observed in iSCAT are essentially averaged out, thus significantly reduces the background noise.

Based on the discussion above, I do not think the work reported in the manuscript is suitable for publishing in Nature Methods considering the limited scope, too much technical details and the lack of significant improvement on the state-of-the-art techniques. The current work is more suitable for more specialized journals such as ACS Nano.

Minor critics:

Line 81-82: "In this arrangement, the nanochannels improve the optical contrast of the imaged biomolecules by several orders of magnitude". What technique are the authors comparing with here with regards to improvement on contrast? Through what mechanism was the contrast improved? Small channel scattering cross section thus small reference field?

Reviewer #3:

Remarks to the Author:

The authors build on their expertise of conducting biological analysis inside nanofluidic channels to present a new imaging method termed Nanofluidic Scattering Microscopy (NSM). I fully agree with the rationale for the method which does not require surface affinity - this difference sets it firmly apart from other existing methods, most notably iScat, and makes the method very powerful. In this regard, the technique is very elegant and I particularly like the fact that it collects more information, namely a) optical contrast AND hydrodynamic radius and b) it tracks molecules in free motion, so can collect information longer than a comparable surface affinity technique.

Overall, this is a great paper and I congratulate the authors on their excellent work!

I only have 2 minor questions that I would like the authors to address;

a) What determines the optimum channel size? I would appreciate a discussion of method performance vs channel size. Related, are there any issues with clogging?

b) Please discuss the limits of the method in terms of analyte concentration and mixed populations of particles/molecules.

Author Rebuttal to Initial comments

Reviewer # 1:

This manuscript presents nanofluidic scattering microscopy (NSM), a label-free approach for solution phase single molecule detection in a confined nanochannel, providing the information about the molecular mass, size, and diffusivity, based on interference scattering imaging. This is an interesting work, as the use of nano channel is a clever way for stable imaging and tracking single molecules in solution phase, which is distinct from most other label-free optical

single molecule detection methods that can only measure molecules attached to a surface. The capability to measure the diffusion of the single molecule in the solution could be used to explore the molecular behavior in real biofluid (Nat Methods 17, 524-530 (2020)). The manuscript is well written in general. However, this work is lack of solid experimental demonstration for the claimed applications and clarification on its limitations. Thus, the manuscript is not ready for publication in current form. I have the following questions and suggestions.

Our reply: We thank the Reviewer for the careful assessment of our work and the insightful and constructive comments, which we address in detail in our point-to-point response below.

1. What is the throughput of the detection? Compared to the wide field imaging approaches (Science 27 Apr 2018: Vol. 360, Issue 6387, pp. 423-427; Nat Methods 17, 1010-1017 (2020)), NSM relies on the confinement of nano channels. Trapping and measure one molecule a time in a nanochannel seems a low throughput. This could be a major limitation of the method. For some of the claimed potential applications, throughput is critical. For example, study of molecular interactions and the heterogeneities of molecular behavior and functions need adequate throughput to get statistically valid result. Also, low-throughput solution phase single molecule binding has been realized with optical tweezer (Analyst, 2015, 140, 4760-4778; Nano Lett. 2014, 14, 10, 5787-5791; ACS Photonics 2014, 1, 5, 389-393).

Our reply: We thank the reviewer for this valuable comment and we agree that our current NSM instrumentation has a throughput that is limited by the volume of a single nanochannel. However, this throughput can be substantially increased (by up to two orders of magnitude) by applying the parallel analysis of a large number of nanochannels that are all connected to the same fluidic system, as well as imaged and analyzed from the same field of view of the microscope. To enable this highly parallelized version of NSM, a CMOS camera with fast read-out (on the order of thousand frames per second) from the whole CMOS chip area is required. This kind of performance is, for instance, offered by the pco.dimax camera that was recently introduced on the market and offers unprecedented performance in terms of framerate. Therefore, we are confident that the limited throughput can be alleviated by using parallelized readout from an array of nanochannels in combination with a state-of-the-art CMOS camera. A second important aspect that can significantly improve the throughput of NSM is the introduction of a constant flow through the nanochannel rather than relying on diffusion alone. In this way, a larger sample volume can be "scanned" in shorter period of time and molecules flowing through the field of view can be tracked and analyzed. A similar concept has been introduced, for instance, in the context of the high-throughput fluorescence-based analysis of exosomes inside nanochannels where up to 2500 particles per minute has been analyzed [1].

We will add a corresponding comment in the discussion/outlook section of the revised manuscript.

2. The behavior of the molecule in a nanochannel is different than in a bulk solution phase due to the surface effect (ACS Omega 2019, 4, 16, 17016-17030; J Phys. Chem. B 2006, 110, 22,

10910-10918). A discussion should be given on how this difference will impact the anticipated applications of the method. For example, how much effect does the interaction between the channel wall and molecule on the measured diffusion coefficient?

Our reply: The diffusivity of a molecule inside a nanochannel can be indeed different from that in bulk because it can be affected by the nanochannel size and geometry. In our analysis, we account for such surface effects by introducing a hindrance factor K_{in} Eq. 2, whose magnitude is estimated using the phenomenological model suggested by Dechadilok et al. [2]. Specifically, we demonstrate this behavior in Fig. 3F by plotting the theoretical dependency of diffusivity on both molecule and nanochannel size. To clarify the difference between diffusivity inside the nanochannel and bulk, we will in the revised manuscript add theoretical values of diffusivity in bulk to Fig. 3F for comparison and explain our analysis in more detail.

As a second aspect, we agree that the diffusivity of a biomolecule inside the nanochannel could be affected by other factors that are not included in the used theory, e.g., electrostatic and hydrophobic interactions of the molecule with the wall, the shape of the molecule, or even that in extremely confined nanochannels the viscosity of the water may be modified. Importantly however, we do not observe any significant contribution from such potential effects in our data since our estimated hydrodynamic radii correspond very well with the values measured using electrophoresis in the bulk (Fig. 3F). Furthermore, an alternative way to address this issue and precisely measure hydrodynamic radii without the use of any theoretical assumptions is to calibrate the system using a calibration set of different molecules (particles) with well-known properties, as for any other analytical tool. Therefore, we are confident that confinement effects are not significant in our system and the results obtained with it. Nevertheless, to clarify this point we will add a brief discussion to the revised manuscript.

3. Single molecule detection easily measures the false signals due to its high sensitivity, so a cross validation method should be provided to confirm the result. Such as the single molecule TIRF used by iSCAT (Nat Commun 5, 4495 (2014)), and antibody recognition used by plasmonic scattering imaging (Nat Methods 17, 1010-1017 (2020)).

We will provide cross-validation experiments based on fluorescence microscopy of DNA molecules. In addition, we will add the analysis of the DNA molecules using electrophoresis, provided by the producer of the samples (ThermoFisher scientific).

4. Single molecules usually have high diffusion rate (Nat Methods 17, 524-530 (2020)). How to ensure the tracked molecule is the same molecule over time in the channel?

Our reply: This is an interesting question asked by the Reviewer. The length of a nanochannel extends the field of view of our microscope. Therefore, the same molecule can, in principle, leave and reenter the field of the view multiple times, creating multiple separate trajectories that cannot be rigorously linked together and assigned to the same molecule. However, the aim of the presented analysis was to determine properties (molecular weight and hydrodynamic radius) of populations of molecules present in the sample and their (relative) concentration. This information is contained in the histograms of the responses (e.g. Fig.3 C, D) where each trajectory is presented by N values of the measured property of every single trajectory where

N is the temporal length of each trajectory (number of frames). (This description is however missing in the current version of the paper and will be added for clarification.) In other words, the analysis does not count number of molecules but rather determine concentration. In that case, the same result is obtained independent from whether a single molecule is presented by multiple short trajectories or one long one.

On a separate note, we highlight that our concept also offers specific advantages with respect to ensuring that a tracked molecule is the same over time since, once in the channel and if a flow is applied through the channel, a molecule cannot leave the field of view as it essentially moves in 1D only and is enforced to remain in the focal plane. In alternative methods (e.g. iSCA T) where molecules in principle are free to move in 3D they can leave the field of view by surface diffusion or transiently leave the focal plane into the bulk solution, thereby making impossible to be tracked over time.

5. Is shot noise the dominant noise in the system as predicted by the calculation in the experimental setup section? A noise analysis similar to Fig. 2e in Nat Commun 5, 4495 (2014) will answer this question.

Our reply: The noise in the system is indeed predominantly shot noise. As suggested by the Reviewer, we will add Fig. Sx in the SI to directly compare the calculated level of the shot noise with the experimental data.

6. The diffusion analysis relies on the localization, and the localization error is related to both signal to noise ratio and objective NA. Thus, a clear discussion about signal to noise ratio is needed. In addition, according to Fig. SJ, the back aperture of objective is blocked partially. This will affect the effective NA (ACS Photonics 2017, 4, 2, 211-216; PNAS September 14, 2010107 (37) 16028-16032). What is the effective objective NA in this optical configuration?

Our reply: To account for localization error in the diffusion analysis, we use the covariance-based estimator suggested by Vestergaard et al. [3] that can be calculated in an unbiased way directly from a series of space displacements determined from every single trajectory (described in Methods-Particle tracking, Equation 13). The second factor in Equation 13 corrects not only for the localization error but also for the motion blur. In other words, in order to correctly determine the diffusivity from the trajectories, the information about the signal-to-noise ratio or localization error is not needed in our case. However, we agree that the information about the localization error is relevant for estimation of the theoretical limits of NSM performance and we will add the related analysis as a part of the section in SI - Resolution in molecular weight and hydrodynamic radius.

In addition, we will add the information about the effective NA in the Methods - Experimental Setup section, as suggested by the Reviewer.

7. Interferometric image contrast for one molecule is determined by the phase. Different distance between the object and interference signal will lead to varying image contrast (Nat Methods 17, 1 OJ 0-1017 (2020)). As shown in the FDTD simulation results (Figure SJ I C and D), the scattering cross section of the biomolecules depends on the location of the molecule in the channel. Are the measured biomolecule signal intensities also showing similar pattern of

location dependency? Is this location dependent intensity corrected for the determination of the molecular weight?

Our reply: Our current temporal resolution (time step is 5 ms) is not low enough to resolve these temporal fluctuations. The molecules that we have imaged have diffusivities 20 - 50 $\mu\text{m}^2/\text{s}$. The mean path of a molecule in one time step then corresponds to about 450 - 700 nm (mean path² = 2 x diffusivity x time step), which is much larger than the nanochannel cross section. This means that even if the difference in the phase contrast would be detectable, every frame of a kymograph represents an image of a molecule averaged over all the axial positions inside the nanochannel. To address this point, we will add a short comment in the revised manuscript. Regarding the determination of the molecular weight from the integrated optical contrast, we do not take into account any location dependency, since the mean value over all possible location inside the nanochannel is approximately equal to the value determined from the analytical theory (Equation 10 and 11 in SI, Figure SI 1 B).

8. For the claimed NSM mass resolution in page 11, can a direct experimental proof be provided similar to iSCAT, by measure molecules with known added weights by modifying different numbers of biotins? (Science 27 Apr 2018: Vol. 360, Issue 6387, pp. 423-427)
To support the claimed resolution of NSM, we will add experimental evidence of antigenantibody binding of molecules with known molecular weight.

9. Could the input and output for each step in the machine learning method (in Methods or in FigS7-9) be stated with determined size and 1D/2D profile? Can some intermediate results be given to help the reader understand the workflow?

Our reply: To address the first point of the reviewer, we will add the requested information to the SL To address the second point, these intermediary steps are all already included in Fig. S6. We will add a clarifying comment to the SI text to make this clearer.

10. If YOLO provides positions and intensities as its output, is FCNN part redundant?

Our reply: The YOLO's output contains no information regarding the intensity of trajectories, and it is not clear where that impression might have been formed in our text.

11. Experimental demonstrations are missing for the claimed applications in the conclusions. These applications are hot topics, but they are non-trivial. For real biofluid detection, good surface chemistry is needed to discriminate the objects from impurities (ACS Sens. 2021, 6, 3, 1357-1366). For deep learning study, both the dataset and the theoretical model should be established well (Nat Methods 18, 194-202 (2021)). Can the authors pick up one of their claims to perform an example experiment? This is important to show the advantage of NSM

Our reply: As the key aspect for most of the discussed future applications of NSM discussed in the outlook section, we suggest to employ established surface modification methods in order to enable NSM analysis of a wide range of different types of molecules, e.g., in terms of charge, as well as in more physiological conditions. Therefore, to follow the Reviewer's request for performing an example experiment, we will include an explicit demonstration of nanochannel

surface modification and the thereby obtained possibility to also study positively charged molecules

with NSM. In addition, we will add analysis of a protein ladder that contains both positively and negatively charged proteins.

In order to illustrate NSMs potential for experiments in realistic biofluids for the sake of this revision, we share here a preliminary result of the analysis of the extracellular vesicles (exosomes) extracted directly from cell culture media. In order to prevent binding of the exosomes to the nanochannel wall and subsequent rapturing, the surface of the nanochannels was passivated by a lipid bilayer formed by surface-induced vesicle rupture [4]. In order to increase the throughput (up to about ten vesicles per minute), a slow flow inside the nanochannels was established. The figure below shows a time-sequence of NSM images (kymograph) revealing the movement (combination of flow and diffusion) of several exosomes inside the nanochannel with cross-sectional area $A = 200 \times 200 \text{ nm}^2$. We note here that the different optical contrast between the captured trajectories indicates different sizes (molecular weights) of the exosomes, which is in good agreement with the typical high heterogeneity of exosomes in a native sample.

To put these preliminary data into a wider context, we want to note that extracellular vesicles recently have attracted substantial attention due to their importance in intercellular communication, as well as in the pathobiology of several diseases [5]. Therefore, they have, for example, been suggested as promising diagnostic biomarker [6] and there is an urgent need for new methods that would enable precise analysis of their size and composition. This is, however, technically highly challenging, due to a high degree of heterogeneity in exosome populations and due to their small sizes (typically from tens to hundreds of nanometers). NSM has a high potential to provide the necessary information about size and molecular weight from which additional information about the composition can be extracted. This is information that is not accessible by any other available technique today, to the best of our knowledge. At the same time, we feel that this topic is highly complex and also out of the scope of the current paper. Therefore, we would strongly prefer to not add any of these results to this paper but only use it for the sake of discussion and to clearly further demonstrate the potential of NSM to address highly relevant biological questions in relevant biological environments.

12. Page 19, line 1, N_s is not defined.

Our reply: We have corrected this mistake and added the definition of N_s as " ... where N_t and N_s are the number of time frames and number of pixels".

Reviewer #2:

The manuscript presents an interferometric scattering imaging modality (Nanofluidic Scattering Microscopy, NSM) in which nanochannels are used to (1) confine the molecules of interest and (2) provide reference light field for interference. In comparison with conventional iSCAT techniques, the authors argue that NSM the major benefits of NSM are: (1) no need for surface attachment, and (2) prolonged sampling time. The authors claim that with these benefits, NSM can achieve 3-fold improvement on molecular weight resolution (30 kDa) comparing with the state-of-the-art iSCAT techniques.

However, based on the theory and data shown, I am not at all convinced that the authors' claims can be supported. To begin with, the authors claim that for iSCAT, "as key point, these three label-free optical single-molecule detection methods require the investigated species to bind to a surface to be 'visible'" (line 52-54), which is not true: In 2019, Taylor R W et al demonstrated via iSCAT the visualization of microsecond nanoscopic protein motion on a live cell membrane with high precision localization, in which the focal plane was placed "at a distance of several micrometres above the glass interface" (Nature Photonics 13. 7 (2019): 480-487). In the manuscript, the authors confine the molecules of interest using nanochannels of different cross sections (Channel I: 100 nm x 27 nm, Channel II: 100 nm x 72 nm, Channel III: 100 nm x 15 nm, Channel IV: 145 nm x 27 nm, Fig. S3). The largest channel height is only 72 nm whereas the smallest channel height can be as small as 15 nm. In comparison, Taylor R W et al demonstrated that iSCAT can achieve an axial localization precision of 4-6 nm for particles within ~ 100 nm axial displacement via PSF fitting. In other words, the NSM technique actually need to bring the molecules closer to the surface than conventional iSCAT techniques.

Our reply: While the Reviewer is correct that in the study by Taylor et al. (Nature Photonics 13.7 (2019): 480-487) the focal plane was placed several micrometers above the glass surface, we want to clearly point out that the detection of the "freely moving proteins" in this study is by no means label free! In fact, this work involves the labelling of the proteins with 20 nm gold nanoparticles (GNPs), which in essence is no different from using other well-established labels like fluorescent tags. Therefore, it is in our opinion not correct to compare this work with our NSM method that is truly label free.

Furthermore, we argue that what the authors in Taylor et al. detect are not "freely moving proteins" but rather protein-GNP complexes where the GNP-tags significantly influence the rates of diffusion due to their mass and additional volume. In other words, extracting the information about the hydrodynamic radius is impossible in this case. Similarly, we note that GNPs provide an about one order of magnitude higher optical contrast than biomolecules of the same size due to their significantly higher refractive index, which is why the authors use them to "enable to visualization of virtually invisible objects". However, as the key point here, they do not visualize the biomolecules themselves in this way but rather the GNP - like any other labelbased

method. Therefore, we are still convinced that the main achievement of NSM is the ability to directly visualize biomolecules that diffuse in a liquid environment, without the use of any labels or any attachment to any surface, and that our key claim is correct and has not been demonstrated before.

As a further aspect related to the Reviewer's statement "In other words, the NSM technique actually need to bring the molecules closer to the surface than conventional iSCAT techniques", we note that is an important difference between being close to a surface and to be chemically bound to a surface since the latter potentially (significantly) alters molecular conformation and function. So, even if we bring the biomolecules close to a surface inside a nanochannel, the key difference is still that we do not require any chemical/electrostatic interaction between the molecule and any kind of surface to be able to visualize the molecules. This is the direct consequence of the fact that the nanochannels ensure that the imaged molecules stay within the focal plane, even if they are freely moving. On the other hand, iSCAT has reported label-free detection of moving proteins only for cases where molecules were bound to some surface, e.g. myosin5 on actin filaments [7] or proteins on clean substrate [8]. There are additional limitations and concerns by utilizing the scattered light from nanochannels as the reference beam:

1. Clogging of molecules within the channel: the authors selectively choose negatively charged molecules in the study to prevent binding events on the channel walls. This greatly limits the practicality of NSM since in most study the molecules are not all negatively charged.

Our reply: We will add an experimental demonstration that positively charged proteins can be imaged using NSM in a coated nanochannel.

Furthermore, our preliminary data on exosomes obtained in a complex biofluid demonstrate that clogging is not a mayor issue.

2. Difficulty in adjusting the optical contrast (iOC in the study) in interferometric scattering modality: $iOC = n \cdot \alpha_m / A$. To adjust iOC for small molecules, the only approach in the NSM method is to adjust A, which is the cross-sectional area of the nanochannel in use. In other words, the users would have to redesign and fabricate a customized nanochannel which involves e-beam lithography and thermal oxidation. In iSCAT, this can be simply achieved by swapping the gold dot attenuator of different extinction ratio.

Our reply: The Reviewer is correct that different types of nanofluidics devices would be used to study biological entities with different dimensions/iOC, where A is adjusted for the purpose at hand. While this at first may appear to be a limiting factor, we would argue that it is not since numerous very-well established experimental techniques - with Scanning Tunneling Microscopy (STM) and Atomic Force Microscopy (AFM) maybe being the most prominent ones - actually require regular exchange of a key component. Similarly, widely used bioanalytical tools like SPR or QCM-D require the regular exchange of the used sensors when new experiments (on different analytes) are targeted. Furthermore, today the fabrication of nanofluidic devices is a very well-established technology. Quoting from L. Bocquet: Nanofluidics coming of age, *Science Materials*, 19, 254-256 (2020): "... The fabrication of nanofluidic devices amenable to systematic investigations was indeed a challenging prerequisite hindering the

development of the field. But the domain has since undergone a quantum leap and a general impression from recent papers and conferences is that an exciting period starts for nanofluidics... systems that seemed a distant dream ten years ago recently became a reality. It is now possible to fabricate individual artificial channels with nanometric and even subnanometric size with manifold geometries. "

Regarding the second aspect of the Reviewer's comment "In iSCAT, this can be simply achieved by swapping the gold dot attenuator of different extinction ratio", we are not entirely clear about what the Reviewer means with "gold dot attenuator" but we assume that s/he probably refers to labelling a molecule with a GNP, as in the work by Taylor et. al already invoked in the comment above (Nature Photonics 13.7 (2019): 480-487). While then the statement by the Reviewer is technically correct in terms of changing optical contrast by swapping the attached type of GNP, we again highlight that this strategy in general cannot be considered "label-free" and therefore it is not entirely correct to compare it with NSM. Furthermore, we argue that "simply swapping gold dot attenuator" may not always be as "simple" as it may appear since it every time involves an additional sample-preparation step which by itself may alter the sample. Therefore, in our opinion, it is not necessarily a more efficient solution than using another NSM chip, in particular once they are available "off-the-shelf", like the GNPs used for labelling or SPR and QCM sensors.

As our last comment, it is also again not clear, as already discussed in the comment above, how iSCAT would ensure the localization of a freely diffusing imaged object within the microscope focal plane since in the aforementioned paper the diffusing object (i.e., the GNP-protein complex) is restricted by the presence of the cell membrane and therefore locked near the focal plane. However, this concept cannot be considered as a general approach for imaging of freely diffusing objects, as the Reviewer is doing here.

3. Limited applications. NSM cannot be used for interferometric scattering imaging in 2D or 3D (for example, cell surface imaging), either can NSM be used for freely diffusive molecule tracking (due to the hindrance effect from the restricted volume of the nanochannel) which greatly limits the scope of the NSM technique as well the throughput (since the molecules are required to flow through a nanochannel rather than freely suspended in solution).

Our reply: As a first general response to this comment, we would like to note that the critique raised here is of technical, rather than fundamental nature. In other words, with the appropriate technical development and refinement, as naturally happening with any newly invented technique over time, many of the current issues that seem to limit applications will be overcome.

Simultaneously, we also want to state that it is not realistic to expect that all these limitations are explicitly resolved upon first reporting a new experimental technique, and that so also was not the case with iSCAT that the Reviewer invokes as the key benchmark.

As to a more specific response, we agree with the Reviewer that NSM may not be possible to use for cell surface imaging but to call this a severe limitation is somewhat exaggerated because, if we again take iSCAT as an example, it is clear that these kinds of experiments only constitute a (small) subgroup of applications. When it comes to molecular tracking of free diffusion, the potential impact of the "hindrance effect" and implications of nanofluidics for the

throughput of NSM, we first refer to a similar discussion in response to Reviewer # 1, which we reproduce here for convenience:

As a further aspect of our reply to this comment, we want to highlight that the fact that we use a nanofluidic solution not only comes with potential limitations, as stated by the Reviewer, but that it actually also opens up new possibilities that are unique to nanofluidics and that are the reason for the wide and steadily growing application of nanofluidic solutions in biology, biochemistry and biophysics, which include the imaging of DNA [9], the analysis of exosomes [10], the analysis of the cell secretion [11] or the analysis of the intracellular content of a single cell [12]. In particular in the last examples, the limited volume of a nanochannel even constitutes a critical benefit since it prevents the excessive dilution of the secretome of a single cell and significantly improves mass-transport characteristics, thereby substantially increasing the probability of detection of secreted molecules compared to detection in traditional single-cell microfluidics.

4. Additional costs for each imaging experiment by the nanochannel chips.

Our reply: As also commented above, there are many widely established techniques that heavily rely on consumables, without that limiting their applicability. More technically speaking, the cost for nanochannel chips is much smaller than what one might expect. In fact, we produce today such chips for our in-house use at a cost of less than 10 €/piece including the labor cost of the nanofabrication specialist who makes them, all the user feeds of the nano fabrication lab, etc. This is negligible compared to the main cost associated with any research project. Furthermore, as with any manufacturing, cost scales with volume and hence, as the technique will become more established and commercially available, chips are consumables available at the same or even lower cost than SPR sensors or QCM crystals, for example. Hence, the cost of chips most certainly is not a limitation of NSM.

Additionally, the authors' claim on "3-fold improvement on molecular weight resolution" was based on a publication in 2019 (Science 360.6387 (2018): 423-427). However, for BSA the MW resolution is approximately the same. In addition, the current state-of-the-art iSCAT technique can achieve the same if not smaller resolution, for example, recently tubulin monomers of ~ 50 kDa was detected with a bandwidth of 5 kDa (Journal of molecular biology 432.23 (2020): 6168-6172.). BSA is of similar MW in the manuscript, yet the resolution reported using NSM is 20 kDa, which is much larger than the current state-of-the-art method. To claim a 3-fold improvement on molecular weight resolution, the authors need to compare with more than just one sample. The authors claim that the improvement was attributed to the prolonged sampling time. Yet I am suspecting that, since NSM uses polychromatic light, the speckle patterns commonly observed in iSCAT are essentially averaged out, thus significantly reduces the background noise.

Our reply: We would like to clarify that in the iSCAT study describing the detection of tubulin (Journal of molecular biology 432.23 (2020): 6168-6172.) the authors do not state the achieved resolution of iSCAT (defined as full width at half-maximum of the measured peak). The value 5 kDa was used in the publication in the context of the bandwidth of kernel density estimates

- a smoothing parameter that has been chosen by the authors to present the data. From the presented images showing the distribution of the measured mass (Figure 1e), we can only estimate the width of the peaks to be approximately 20 kDa - comparable to the resolution reported in (Science 360.6387 (2018): 423-427) for similar range of molecular weights. However, we agree that our claim "3-fold improvement on molecular weight resolution" is not accurate - the stated improvement was achieved for a molecule in the range of hundreds of kDa, in the range of tens of kDa the resolution is rather comparable to the state-of-art iSCAT. Therefore, we will modify this statement into "up to 3-fold improvement on molecular weight resolution".

Regarding the reviewer's comment "Yet I am suspecting that, since NSM uses polychromatic light, the speckle patterns commonly observed in iSCAT are essentially averaged out, thus significantly reduces the background noise.", in publications (Science 360.6387 (2018): 423-427) or similar iSCAT study [13] where monochromatic light was used, the authors claim that the iSCAT resolution is limited by photon shot noise, therefore, not influenced by speckle patterns.

Minor critics:

Line 81-82: "In this arrangement, the nanochannels improve the optical contrast of the imaged biomolecules by several orders of magnitude". What technique are the authors comparing with here with regards to improvement on contrast? Through what mechanism was the contrast improved? Small channel scattering cross section thus small reference field?

Our reply: This statement refers to the intensity of the light scattered by a biomolecule alone, diffusing in bulk solution, outside the nanochannel. This statement is then supported and explained by paragraph below this statement and Fig. 1 B-D. To make this clear, we will modify the sentence and add the corresponding reference to Figure 1B-D.

Reviewer #3:

The authors build on their expertise of conducting biological analysis inside nanofluidic channels to present a new imaging method termed Nanofluidic Scattering Microscopy (NSM). I fully agree with the rationale for the method which does not require surface affinity - this difference sets it firmly apart from other existing methods, most notably iScat, and makes the method very powerful. In this regard, the technique is very elegant and I particularly like the fact that it collects more information, namely a) optical contrast AND hydrodynamic radius and b) it tracks molecules in free motion, so can collect information longer than a comparable surface affinity technique.

Overall, this is a great paper and I congratulate the authors on their excellent work!

Our reply: We would like to thank the Reviewer for this very positive assessment of our work and appreciate the congratulations.

I only have 2 minor questions that I would like the authors to address; a) What determines the optimum channel size? I would appreciate a discussion of method performance vs channel size. Related, are there any issues with clogging?

Our reply: We thank the reviewer for a valuable comment since optimization of the channel size is indeed an important topic that has been only briefly mentioned in the context of the

obtained resolution for Channel I and II (SI - Resolution in molecular weight and hydrodynamic radius, Fig. S19). In order to provide a more general picture of the role of nanochannel size, we will add a corresponding analysis of the dependency of NSM resolution on nanochannel size. We will discuss both theoretical limits, as well as experimental observations, including a discussion of clogging issues and how to resolve them.

b) Please discuss the limits of the method in terms of analyte concentration and mixed populations of particles/molecules.

Our reply: The maximum concentration is limited by the volume of the nanochannel that corresponds to the diffraction limited spot and the ability of the particle tracking algorithm to reliably distinguish intersecting trajectories of different molecules. The minimum concentration is mostly limited by the detection time and the shot noise of the system. To explicitly discuss this aspect, we will add a detailed description in the section "Limit of detection" in the revised SI.

Regarding the limits of the method in terms of mixed populations, we will add the results of the analysis of a protein ladder together with the related discussion to our revised work.

References

- [1] R Friedrich et al., A nano flow cytometer for single lipid vesicle analysis. *Lab Chip* 17 (5), 830 (2017).
- [2] P Dechadilok and WM Deen, Hindrance factors for diffusion and convection in pores. *Ind Eng Chem Res* 45 (21), 6953 (2006).
- 11
- [3] CL Vestergaard et al., Optimal estimation of diffusion coefficients from single-particle trajectories. *Phys Rev E* 89 (2) (2014).
- [4] F Persson et al., Lipid-based passivation in nanofluidics. *Nano Lett* 12 (5), 2260 (2012).
- [5] Evan der Pol et al., Classification, functions, and clinical relevance of extracellular vesicles. *Pharmacol Rev* 64 (3), 676 (2012).
- [6] RJ Simpson et al., Exosomes: proteomic insights and diagnostic potential. *Expert Rev Proteomics* 6 (3), 267 (2009).
- [7] J Ortega Arroyo et al., Label-free, all-optical detection, imaging, and tracking of a single protein. *Nano Lett* 14 (4), 2065 (2014).
- [8] M Liebel et al., Ultrasensitive Label-Free Nanosensing and High-Speed Tracking of Single Proteins. *Nano Lett* 17 (2), 1277 (2017).
- [9] V Muller and F Westerlund, Optical DNA mapping in nanofluidic devices: principles and applications. *Lab Chip* 17 (4), 579 (2017).
- [10] R Szatanek et al., The Methods of Choice for Extracellular Vesicles (EVs) Characterization. *Int J Mol Sci* 18 (6) (2017).
- [11] K Shirai et al., Extended Nanofluidic Immunochemical Reaction with Femtoliter Sample Volumes. *Small* 10 (8), 1514 (2014).
- [12] L Lin et al., Micro/extended-nano sampling interface from a living single cell. *Analyst* 142 (10), 1689 (2017).
- [13] M Piliarik and V Sandoghdar, Direct optical sensing of single unlabelled proteins and

super-resolution imaging of their binding sites. Nat Commun 5, 4495 (2014).

Decision Letter, first revision:

Dear Christoph,

Thank you for your letter asking us to reconsider our decision on your Article, "Nanofluidic Scattering Microscopy for Label-free Weight and Size Screening of Single Diffusing Biomolecules". After careful consideration we have decided that we are willing to consider a revised version of your manuscript that adds the additional experiments and clarifications proposed. We also would like you to include in the revision the preliminary data on EV tracking, as we think this represents a challenging demonstration and would broaden the appeal of the paper.

- * include a point-by-point response to our referees and to any editorial suggestions
- * please underline/highlight any additions to the text or areas with other significant changes to facilitate review of the revised manuscript
- * address the points listed described below to conform to our open science requirements
- * ensure it complies with our general format requirements as set out in our guide to authors at www.nature.com/naturemethods
- * resubmit all the necessary files electronically by using the link below to access your home page

[Redacted] URL links to your confidential home page and associated information about manuscripts you may have submitted, or that you are reviewing for us. If you wish to forward this email to co-authors, please delete the link to your homepage.

We hope to receive your revised paper within 2-3 months. If you cannot send it within this time, please let us know. In this event, we will still be happy to reconsider your paper at a later date so long as nothing similar has been accepted for publication at Nature Methods or published elsewhere.

OPEN SCIENCE REQUIREMENTS

REPORTING SUMMARY AND EDITORIAL POLICY CHECKLISTS

When revising your manuscript, please submit reporting summary and editorial policy checklists.

Please note that these forms are dynamic ‘smart pdfs’ and must therefore be downloaded and completed in Adobe Reader. We will then flatten them for ease of use by the reviewers. If you would like to reference the guidance text as you complete the template, please access these flattened versions at <http://www.nature.com/authors/policies/availability.html>.

DATA AVAILABILITY

Please include a “Data availability” subsection in the Online Methods. This section should inform readers about the availability of the data used to support the conclusions of your study, including accession codes to public repositories, references to source data that may be published alongside the paper, unique identifiers such as URLs to data repository entries, or data set DOIs, and any other statement about data availability. At a minimum, you should include the following statement: “The data that support the findings of this study are available from the corresponding author upon request”, describing which data is available upon request and mentioning any restrictions on availability. If DOIs are provided, please include these in the Reference list (authors, title, publisher (repository name), identifier, year). For more guidance on how to write this section please see: <http://www.nature.com/authors/policies/data/data-availability-statements-data-citations.pdf>

CODE AVAILABILITY

Please include a “Code Availability” subsection in the Online Methods which details how your custom code is made available. Only in rare cases (where code is not central to the main conclusions of the paper) is the statement “available upon request” allowed (and reasons should be specified).

For more information on our code sharing policy and requirements, please see: <https://www.nature.com/nature-research/editorial-policies/reporting-standards#availability-of-computer-code>

ORCID

Sincerely,
Rita

Rita Strack, Ph.D.
Senior Editor
Nature Methods

Author Rebuttal, first revision:

Reviewer #1:

This manuscript presents nanofluidic scattering microscopy (NSM), a label-free approach for solution phase single molecule detection in a confined nanochannel, providing the information about the molecular mass, size, and diffusivity, based on interference scattering imaging. This is an interesting work, as the use of nano channel is a clever way for stable imaging and tracking single molecules in solution phase, which is distinct from most other label-free optical single molecule detection methods that can only measure molecules attached to a surface. The capability to measure the diffusion of the single molecule in the solution could be used to explore the molecular behavior in real biofluid (Nat Methods 17, 524–530 (2020)). The manuscript is well written in general. However, this work is lack of solid experimental demonstration for the claimed applications and clarification on its limitations. Thus, the manuscript is not ready for publication in current form. I have the following questions and suggestions.

Our reply: We thank the Reviewer for the careful assessment of our work and the insightful and constructive comments, which we address in detail in our point-to-point response below.

1. What is the throughput of the detection? Compared to the wide field imaging approaches (Science 27 Apr 2018: Vol. 360, Issue 6387, pp. 423-427; Nat Methods 17, 1010–1017 (2020)), NSM relies on the confinement of nano channels. Trapping and measure one molecule a time in a nanochannel seems a low throughput. This could be a major limitation of the method. For some of the claimed potential applications, throughput is critical. For example, study of molecular interactions and the heterogeneities of molecular behavior and functions need adequate throughput to get statistically valid result. Also, low-throughput solution phase single molecule binding has been realized with optical tweezer (Analyst, 2015, 140, 4760-4778; Nano Lett. 2014, 14, 10, 5787–5791; ACS Photonics 2014, 1, 5, 389–393).

Our reply: We thank the reviewer for this valuable comment and we agree that our current NSM instrumentation has a throughput that is limited by the volume of a single nanochannel. However, this is mitigated by the fact that multiple molecules at the same time can be measured in the same nanochannel, because the length of the illuminated area is long enough to accommodate and discriminate several diffusing molecules. In addition, the NSM throughput can be increased by up to two orders of magnitude by performing a parallel analysis of a large number of nanochannels, which are all connected to the same fluidic system while being imaged and analyzed using the same field of view of the microscope. To enable this highly parallelized version of NSM, a CMOS camera with fast read-out (on the order of thousand frames per second) from the whole CMOS chip area is required. This kind of performance is, for instance, offered by the pco.dimax camera that was recently introduced on the market and offers unprecedented performance in terms of framerate. This is therefore certainly a direction of development that we will consider.

A second important aspect that can significantly improve the throughput of NSM is the introduction of a constant fluid flow through the nanochannel. Doing so employs directed transport to overcome the limitations imposed by relying on diffusion alone. In this way, a larger sample volume can be “scanned” in shorter period of time and molecules flowing through the field of view can be tracked and analyzed. A similar concept has been introduced, for instance, in the context of the high-throughput fluorescence-based analysis of exosomes inside nanochannels where up to 2500 particles per minute have been analyzed (Lab Chip 17, 830-841, (2017)).

As part of the revision of the manuscript, we have applied this strategy to study extracellular vesicles in cell culture medium. This new material is now added as a new section to the revised manuscript (details can be found the response to the reviewer's comment 11).

Specifically, in order to comment on the throughput, the following text has been added in the revised manuscript in the new section on extracellular vesicles:

"To increase the throughput to up to 10 particles per minute, we introduced a slow flow by applying a 0.25 Pa pressure drop along the nanochannel, resulting in individual particles being flushed through it, ..."

And the following comment has been added in the Conclusions:

"... Furthermore, rapid progress in detector technologies and machine-learning-based data treatment promises to push the limits of NMS towards even smaller molecules and to enable up to two orders of magnitude increased throughput via parallel analysis of a large number of nanochannels."

In addition, to broaden the description of the state-of-art single molecule imaging methods, the reference noted by the reviewer (Nat Methods 17, 1010–1017 (2020)) that describes plasmonic approach based on metallic continuous film, was added to the Introduction:

"Label-free single-biomolecule detection has recently been enabled by dielectric micro-resonators¹, plasmonic approaches based on metallic continuous films² and nanostructures³, and interferometric scattering microscopy (iSCAT)⁴⁻⁸."

2. The behavior of the molecule in a nanochannel is different than in a bulk solution phase due to the surface effect (ACS Omega 2019, 4, 16, 17016–17030; J. Phys. Chem. B 2006, 110, 22, 10910–10918). A discussion should be given on how this difference will impact the anticipated applications of the method. For example, how much effect does the interaction between the channel wall and molecule on the measured diffusion coefficient?

Our reply: The diffusivity of a molecule inside a nanochannel can indeed be different from that in bulk. In our analysis, we account for surface effects by introducing a hindrance factor K in Eq. 2, whose magnitude is estimated using the phenomenological model suggested by Dechadilok et al. 9. Specifically, we demonstrate this behavior in Fig. 3F by plotting the theoretical dependency of diffusivity on both molecule and nanochannel size. To clarify this and explain in more detail the effects that our analysis accounts for, we have modified the manuscript in the following way:

“... by approximating it as a hard neutral sphere, its hydrodynamic (Stokes) radius, R_s , can be estimated using the Stokes-Einstein equation corrected for hindrance effects associated with the diffusion of small objects in a restricted volume as⁹

$$R_s = K \cdot (k_B T) / 6\pi\eta D \quad \text{Equation 2}$$

where k_B is the Boltzmann constant, T is temperature, η is the viscosity of the liquid in the nanochannel, and K is the hindrance factor that takes particle-wall hydrodynamic interactions and steric restrictions inside a nanochannel into account. It is dependent on the size of the nanochannel relative to the dimensions of the biomolecule, and can be estimated using a phenomenological model suggested by Dechadilok et al. 9 as ... “

As a second aspect, we agree that the diffusivity of a biomolecule inside the nanochannel could be slightly affected by other factors that are not included in the used theory. To clarify this point we have added a brief discussion to the revised manuscript:

“We note here that the diffusivity also can be slightly affected by other surface-related effects that are not included here, such as the partial-slip boundary condition¹⁰.”

Importantly however, in the case of protein analysis, we do not observe any significant contribution from such potential effects in our data, as demonstrated by the fact that our estimated hydrodynamic radii correspond very well with the values measured using electrophoresis in the bulk (Fig. 3F). We have added the following note to the text:

“We find that the theoretically predicted values reproduce the experimentally measured D values well, which suggests that potential surface-related effects are negligible.”

In addition, in the case of the added analysis of extracellular vesicles in a conditioned cell culture medium, we have estimated the contribution of the partial-slip boundary condition to be less than 4%, based on published theory¹⁰. The details of the calculation are summarized in the new SI Section 5 – “Error estimate of size determination due to partial-slip boundary condition”.

3. Single molecule detection easily measures the false signals due to its high sensitivity, so a cross validation method should be provided to confirm the result. Such as the single molecule TIRF used by iSCAT (Nat Commun 5, 4495 (2014)), and antibody recognition used by plasmonic scattering imaging (Nat Methods 17, 1010–1017 (2020)).

Our reply: We agree with the reviewer that highly sensitive methods often have a tendency to also generate false signals and that it therefore is important to explicitly address this issue in our work. Therefore, to explicitly demonstrate NSM's ability to resolve molecules and to distinguish them from potential false signals, we have performed an analysis of a "blank sample" comprised of buffer without any molecules. This analysis clearly shows that any signal resembling molecules of molecular weight > 66kDa (the smallest size investigated in the paper) is completely lacking for a "blank sample". Therefore, we feel confident that any signal detected in the corresponding iOC range is real and not a false signal. To discuss this important point explicitly in the revised manuscript, we have added following text to the SI section 6 – "Evaluation of the data processing".

"We also highlight that none of these tests revealed any false signals — i.e., no trajectories other than those simulated were found by the particle tracking algorithm. In addition, analysis of experimentally recorded background signal from a channel filled with pure PBS buffer, without the target molecules, did not reveal any false signal either (Fig. S16). That is the consequence of the fact that the used particle tracking algorithm discards all trajectories that have lower signal than 4 STD of the noise. In the case of machine learning, the models are trained on a wide variation of simulated noise distributions and train-validated against experimental channel noise to ensure that they do not find false signals in the experimentally relevant conditions that they have been validated against. Furthermore, we highlight that if the experimental conditions and noise distribution were to substantially change, the model could quickly be retrained by transfer-learning using the first few seconds of measurement of an empty channel and thereby always prevent false signal detection.

Fig. S16. Example of kymograph corresponding to the experimentally recorded background signal from a nanochannel filled with pure PBS buffer, without the target molecules. When analysing this kymograph,

correctly neither the standard algorithm nor the machine-learning algorithm identify a molecule trajectory.”

Regarding the cross validation suggested by the reviewer, we note that the architecture of the NSM fluidic chip is not compatible with the requirements for the suggested single-molecule approaches. Specifically, the nanochannels are not placed near any interface that would provide total internal reflection and do not contain additional metallic layers to enable plasmonic scattering imaging. Even though both approaches are in principle possible to integrate into our current chip and combine with NSM readout, both would require very substantial changes in the architecture of the chip that could result in different NSM performance characteristics and we therefore consider them beyond the scope of this work.

4. Single molecules usually have high diffusion rate (Nat Methods 17, 524–530 (2020)). How to ensure the tracked molecule is the same molecule over time in the channel?

Our reply: The scattering signal is continuously recorded, so the positions of a molecule are temporally linked. This proper linking into a continuous trajectory that corresponds to the same molecule is ensured by the particle tracking algorithm. To highlight and clarify this point we have modified the text in the following way:

“... To extract iOC and the position of the biomolecule along the nanochannel, x , we used a particle tracking algorithm (see Methods – “Particle tracking”). It evaluates each frame in the kymograph, finds the responses corresponding to a biomolecule and connects them in a trajectory (Fig. 2B). Each biomolecule is then represented ...”

As a second aspect of this important question raised by the reviewer, the length of a nanochannel extends the field of view of our microscope. Hence, the same molecule can, in principle, leave and reenter the field of the view multiple times, thereby creating multiple separate trajectories that cannot be rigorously linked together and unambiguously assigned to the same molecule. However, the aim of the presented analysis is not to count the exact number of molecules but rather to determine the relative concentration of populations of molecules present in the sample. To clarify this, we have added the following description in the main text:

“... To further analyze the obtained single biomolecule data, we plot one-dimensional histograms of iOC converted to MW for all biomolecules in Channel I and II obtained by SA (Fig. 3C, D – Fig. S4C, D for ML). Each detected trajectory is presented in the histogram by $N/(\sum N)$ counts of its determined iOC, where $\sum N$ is the sum of number of frames of all the trajectories that were identified in the sample. This way, the numbers of counts within the peaks correspond to the (relative) concentration of different populations present in the sample. The positions of the main peaks that correspond to molecular monomers are marked ...”.

On a separate note, we highlight that our concept also offers specific advantages with respect to ensuring that a tracked molecule is the same over time since, once in the channel and if a flow is applied through the channel, a molecule cannot leave the field of view as it essentially moves in 1D, and thus is enforced to remain in the focal plane. In alternative methods (e.g., iSCAT) where molecules in principle are free to move in 3D they can leave the field of view by surface diffusion or transiently leave the focal plane into the bulk solution, thereby making it impossible to track them over extended amounts of time.

5. Is shot noise the dominant noise in the system as predicted by the calculation in the experimental setup section? A noise analysis similar to Fig. 2e in Nat Commun 5, 4495 (2014)) will answer this question.

Our reply: The noise is indeed predominantly shot noise, as shown by the analysis suggested by the Reviewer, now presented in SI section 7 – “Noise analysis”:

“To evaluate the level of the noise in the system, we measured the standard deviation of the intensity in a kymograph corresponding to the background recorded from a nanochannel filled with pure PBS buffer (Fig. S16) over 500 s for a range of time frame averaging values (temporal length of one video frame). The camera was operated at 5400 frames per second. The experimental values are compared with the expected level of shot noise calculated as $1/(\sqrt{N_t N_s N_{(e^-)}})$, where $N_{(e^-)}$ is the well depth of photoelectrons in a single pixel ($N_{(e^-)}=33 \text{ ke}^-$ for the Andor Zyla CMOS camera), N_s corresponds to spatial averaging (the signal was averaged over a diffraction-limited spot, $N_s=20 \times 20$ pixels) and N_t corresponds to the time frame averaging. Fig. S19 shows that the experimental values are very close to the theoretical shot-noise limit, which confirms that the measurement is shot-noise limited.

Fig. S19. Noise analysis. Comparison of the experimentally measured standard deviation of the noise with the theoretical limit of the shot noise for a range of time averaging (temporal length of the video frame). The experimental values are very close to the theoretical limit, which confirms that the measurement is shot-noise limited.”

6. The diffusion analysis relies on the localization, and the localization error is related to both signal to noise ratio and objective NA. Thus, a clear discussion about signal to noise ratio is needed. In addition, according to Fig. S1, the back aperture of objective is blocked partially. This will affect the effective NA (ACS Photonics 2017, 4, 2, 211–216; PNAS September 14, 2010 107 (37) 16028-16032). What is the effective objective NA in this optical configuration?

Our reply: To account for localization errors in the diffusion analysis, we use the covariance-based estimator suggested by Vestergaard et al. 11 that can be calculated in an unbiased way directly from a series of space displacements determined from every single trajectory, described in Methods – “Particle tracking”, Equation 13, reported here for convenience:

$$D = \frac{(\Delta x_n)^2}{2\Delta t} + \frac{(\Delta x_n - \Delta x_{n+1})^2}{\Delta t}. \quad \text{Equation 13}$$

The second factor in Equation 13 corrects not only for the localization error but also for the motion blur. Therefore, in order to determine the diffusivity from the trajectories, the information about the signal-to-noise ratio, objective NA (size of the diffraction limited spot) or localization error is not needed in our case.

In addition, we have added the information about the effective NA in the Methods – “Experimental Setup” section, as suggested by the Reviewer:

“The effective NA of the objective was then lowered to 1.27 as the back aperture of the objective was partially blocked — decreased from a diameter of 23.8 mm to 16.7 mm.”

7. Interferometric image contrast for one molecule is determined by the phase. Different distance between the object and interference signal will lead to varying image contrast (Nat Methods 17, 1010–1017 (2020)). As shown in the FDTD simulation results (Figure S11 C and D), the scattering cross section of the biomolecules depends on the location of the molecule in the channel. Are the measured biomolecule signal intensities also showing similar pattern of location dependency? Is this location dependent intensity corrected for the determination of the molecular weight?

Our reply: Our current temporal resolution (time step is 5 ms) is not low enough to resolve these temporal fluctuations. The molecules that we have imaged have diffusivities 20 – 50 $\mu\text{m}^2/\text{s}$. The mean path of a molecule in one time step then corresponds to about 450 – 700 nm (mean path = $(2 \times \text{diffusivity} \times \text{time step})^{1/2}$), which is much larger than the nanochannel cross section. This means that even if the difference in the phase contrast would be detectable, every frame of a kymograph represents an image of a molecule averaged over all the axial positions inside the nanochannel.

Regarding the determination of the molecular weight from the integrated optical contrast, we do not take into account any location dependency, since the mean value over all possible location inside the nanochannel is approximately equal to the value determined from the analytical theory (Equation 10 and 11 in SI, Fig. S12 B).

To address this point, we have added the mean scattering cross section values into Fig. S12B and added the following comment in the revised SI:

“The overall trend captured by both approaches is in excellent quantitative agreement (Fig. S2A). However, the FDTD results show an additional dependency on the axial position of the biomolecule inside the nanochannel that is neglected by the analytical model and that originates from the phase difference between the light scattered by the nanochannel and the biomolecule (Fig. S11B, C, D). As a result, when a biomolecule freely diffuses inside the nanochannel, the scattering cross section randomly fluctuates over time, with the frequency depending on the diffusivity of the molecule (D). However, the mean value over all possible locations is very close to the analytical model (Fig. S11B). This means that if the mean travel distance, $\sqrt{2D\Delta t}$, within one integration time step (Δt) is much larger than the dimensions of the nanochannel cross section, the phase differences are effectively averaged, and the obtained scattering cross section value corresponds very well to the analytical model.

Fig. S11. Simulated relative difference between the scattering cross section of a nanochannel containing a biomolecule, and an empty nanochannel — $((\sigma_t - \sigma_c)) / \sigma_c$. A calculation using an analytical model (Equation 10 and Equation 11) is compared with FDTD simulations. The insets in the figures show schematics of the different calculated/simulated scenarios in a cross-sectional view. (A) Dependency of $((\sigma_t - \sigma_c)) / \sigma_c$ on the linear density of biomolecules ($n_L = N/L$) for a nanochannel with radius $r_c = 50$ nm for TM and TE polarization and for a range of biomolecules defined by their radius (r_p). The

biomolecules were assumed to be in the centre of the nanochannel. (B) Dependency of $((\sigma_t - \sigma_c)) / \sigma_c$ on the radius of a nanochannel (r_c) for TM and TE polarization and a biomolecule with $r_p = 3.6$ nm and $nL = 10 \mu\text{m}^{-1}$ inside for different geometric scenarios with respect to the position of the molecule inside the channel (see inset): (i) in the centre, (ii) at the top, (iii) at the side and (iv) averaged over all possible positions. (C, D) Dependency of $((\sigma_t - \sigma_c)) / \sigma_c$ on the (C) horizontal and (D) vertical position of a biomolecule with $r_p = 3.6$ nm and $nL = 0.1 \mu\text{m}^{-1}$ inside a nanochannel with $r_c = 100$ nm for TM and TE polarization.

8. For the claimed NSM mass resolution in page 11, can a direct experimental proof be provided similar to iSCAT, by measure molecules with known added weights by modifying different numbers of biotins? (Science 27 Apr 2018: Vol. 360, Issue 6387, pp. 423-427)

Our reply: Measurement of the added weights to a biomolecule is, in principle, possible by NSM but we believe that the claimed resolution is already sufficiently supported by the experiments provided by the current study. The claimed NSM mass resolution is determined in the same way as for any other analytical techniques that measure molecular mass (e.g., iSCAT, Science 27 Apr 2018: Vol. 360, Issue 6387, pp. 423-427), i.e., as a full-width-at-half-maximum of the peak corresponding to the same type of molecule. In addition, the claimed experimental values are validated by the detailed analysis that shows that the claimed resolution is in line with the expectations set by theoretical limits (SI section 8 – “Resolution in molecular weight and hydrodynamic radius”).

9. Could the input and output for each step in the machine learning method (in Methods or in FigS7-9) be stated with determined size and 1D/2D profile? Can some intermediate results be given to help the reader understand the workflow?

Our reply: To address the first point of the reviewer, we have added the requested information:

in the end of Methods – “Image Segmentation with U-Net”: “The input data during training are simulated images (kymographs) of size 128×2048 with a random number of trajectories and of output segmented images of equivalent size. The model is also train-validated every 120 epochs (≈ 30000 simulated kymographs) against 150 simulated kymographs (of size 128×512 , 128×1024 , 128×2048) with experimentally measured channel noise, using an 80-20 train-validation split.”

in the end of Methods – “Trajectory Identification with YOLOv3”: “The input data during training are simulated images (segmented kymographs) of size 128×8192 , down-sampled to 128×128 to improve

performance, and the output is a list of YOLO-labels containing class, position, occurrence probability and class probability of each trajectory in the input image.”

in the end of Methods – “Property Calculation with FCNN”: “The input during training of the FCNNs are simulated images (kymographs) of size 128×2048 with a single particle trajectory, and the output is a single value of either iOC or D of said trajectory, as well as the mask, which is a learnt downsampled representation of the original kymograph.”

To address the second point, these intermediary results are now included in Fig. S6. For further clarity, we have emphasized this in Methods – “Machine learning (ML) analysis” with the following comments: “All intermediate results of the ML algorithm are described in the sub-sections below, and the whole pipeline is summarized in Fig. S6.”

10. If YOLO provides positions and intensities as its output, is FCNN part redundant?

Our reply: The YOLO's output contains no information regarding the intensity of the trajectories. We have now clarified this in the revised text, as described in the response to the previous comment.

11. Experimental demonstrations are missing for the claimed applications in the conclusions. These applications are hot topics, but they are non-trivial. For real biofluid detection, good surface chemistry is needed to discriminate the objects from impurities (ACS Sens. 2021, 6, 3, 1357–1366). For deep learning study, both the dataset and the theoretical model should be established well (Nat Methods 18, 194–202 (2021)). Can the authors pick up one of their claims to perform an example experiment? This is important to show the advantage of NSM.

Our reply: We agree with the reviewer that a specific example experiment would strengthen our work even further and we have therefore opted to do so — which is the main reason why this revision has taken us so long time since we had to reach out to external collaborators. Specifically, to explicitly illustrate the capabilities of NSM for experiments with real biofluids, we have performed experiments to detect and quantitatively analyze extracellular vesicles (exosomes) from conditioned cell culture medium. This, together with a second example of the quantitative analysis of the positively charged

protein aldolase, was enabled by coating the nanochannel surfaces with a supported lipid bilayer, which therefore also constitutes an example for specifically tailored surface chemistry, as requested by the reviewer. The text, figures and discussion added to the revised work to accommodate these new examples are included below for convenience.

Regarding “For deep learning study, both the dataset and the theoretical model should be established well (Nat Methods 18, 194–202 (2021)), we note that, differently from the study mentioned by the referee (Nat Methods 18, 194–202 (2021)), our work is not a machine-learning-focused study: i.e., it is not focused on demonstrating that a machine-learning method outperforms a standard method of analysis, which would permit to establish model and dataset. Instead, we use a machine-learning method to replicate and validate the results of a mathematically based algorithm, which we also introduce in this article, and thereafter apply it in regimes which are difficult to tackle with a mathematical approach alone.

Below we summarize all the additional new experiments performed and report the corresponding discussions we have included in the revised version of the manuscript and SI.

1. The following section has been added to the main text, describing the passivation of the nanochannel walls and the analysis of positively charged molecule aldolase:

Surface passivation by supported lipid bilayer

So far, we had designed our experiments such that the biomolecules were predominantly negatively charged to minimize attractive interaction – and thus nonspecific binding – to the negatively charged nanochannel walls. Nevertheless, we were able to observe rare events of molecular binding and unbinding via electrostatic interactions (Fig. S5). For the data presented above we excluded these transient binding events from the analysis (see Methods – “Particle tracking”). At the same time, we also note that the demonstrated observation of nonspecific binding events opens the door to utilizing NSM for affinity-based single molecule detection by analyte-specific receptors immobilized on nanochannel walls¹².

To now demonstrate the active prevention of non-specific binding, we applied a supported lipid bilayer (SLB) coating¹³ on the nanochannel walls, which is formed by adsorption and subsequent rupturing of large unilamellar vesicles (LUVs) (Fig. 4A, details in Methods – “Lipid bilayer”). The real-time NSM-response to the SLB formation inside a 225 × 200 nm² nanochannel (Channel V, Fig. S3E) is shown in Fig.

4B. To demonstrate the surface-passivation effect of the SLB, we also coated a $82 \times 40 \text{ nm}^2$ nanochannel (Channel VI, Fig. S3F) and used it to successfully track the positively charged protein aldolase, which was impossible to analyze in an uncoated channel due to the strong electrostatic interaction with the negatively charged nanochannel walls (Fig. 4C). The obtained MW and RS values for aldolase are in good agreement with the nominal values (Fig. 4D,E), and thus corroborate the wide applicability of NSM, irrespective of analyte charge.

Fig. 4. Surface passivation. (A) Schematic of SLB formation. LUVs flow through the nano-channel, adsorb on the nanochannel wall, rupture and create patches of lipids that eventually connect and create a homogenous layer. (B) Kymograph capturing the SLB formation in Channel V, as manifested by decrease of scattering intensity. (C) Schematic of a biomolecule diffusing inside a nanochannel coated with an SLB. (D) Inferred molecular weight and (E) hydrodynamic radius of the positively charged protein aldolase measured in Channel VI and analyzed by ML. The arrows indicate the nominal values (MW = 158 kDa, RS = 4.6 nm).

2. The following section has been added to the main text of the manuscript, describing the biofluid analysis:

Analysis of extracellular vesicles in conditioned cell culture medium

EVs act as mediators of physiological intercellular communication, play key roles in the pathobiology of several diseases¹⁵ and are promising diagnostic biomarkers¹⁶. Their functionality depends on both their composition and their size. However, these parameters are challenging to precisely define with existing methodologies, such as Dynamic Light Scattering or Nanoparticle Tracking Analysis, due to the significant heterogeneity of EVs, in combination with their small sizes down to the tens of nanometers range¹⁷, and their existence in complex biofluids.

Here, to apply NSM to complex biological sample analysis, we collected conditioned medium from human SH-SY5Y cells (details in Methods – “Conditioned cell culture medium”), containing a mixture of serum proteins and secreted EVs. To accommodate the size of EVs, we used a 225×200 nm² nanochannel (Channel V), whose walls were passivated with an SLB to prevent nonspecific binding. To increase the throughput to up to 10 particles per minute, we introduced a slow flow by applying a 0.25 Pa pressure drop along the nanochannel, resulting in individual particles being flushed through it, as revealed by the NSM signal (Fig. 5A).

Using this setup, we then collected a significant number of trajectories of different BNPs, from which we derive *iOC* and *D* translated to *RS* using Equation 2 (the applicability of the no-slip condition is discussed in SI section 5). This analysis reveals two distinct populations (Fig. 5B): one in the small *iOC/RS* regime (red circle), and one in the large *iOC/RS* regime (blue circle). A control experiment with cell culture medium that had not been in contact with the cells reveals that the former population can be attributed to entities present in the serum supplement, likely lipoprotein particles (Fig. 5C), whereas the second population indeed corresponds to EVs secreted by the cells. The identified *RS* of 10 – 13 nm and 20 – 70 nm (Fig. 5D,E) are in very good agreement with values reported for low-density lipoproteins¹⁸ and EVs¹⁷, respectively, and thus demonstrate the applicability of NSM to detect and, importantly, distinguish non-labelled analytes of biological relevance in a complex sample mixture retrieved from cell culture. However, we also note that precise translation of *iOC* into *MW* is more complicated for BNPs than for single molecules due to a large variety of molecular constituents with different optical properties whose representation and spatial distribution might be different for each BNP (i.e., different constant *a* in Equation 1). Therefore, we did not further explore the content of the EVs via the measured *iOC*.

Fig. 5. Analysis of BNPs in conditioned cell culture medium containing serum. (A) Kymo-graph of multiple lipoprotein particles (marked by red arrows) and one larger EV (marked by the blue arrow) moving through the nanochannel. (inset) Schematics of an EV and a lipoprotein (depicted to scale). (B,C) Scatter plots of *iOC* and *D* translated into *RS*. (D,E) Histograms of *RS*, analyzed using ML and corresponding to (B,D) the conditioned SH-SY5Y human cell medium and (C,E) the control where the medium had not

been in contact with the cells. All data were acquired in Channel V that had been passivated by an SLB prior to the measurement.

3. Since different nanochannels have been used for the analysis of the cell culture medium (Channel V) and positively charged proteins (Channel VI), SEM images of the cross sections of these additional nanochannels have been added as Fig. S3E,F to the SI.

Fig. S3. SEM images of nanochannel cross sections of (A-F) Channel I – VI used in different experiments. The domain from which the area of the cross section was calculated is highlighted by the dashed line. The width and length correspond to the dimensions of a rectangle with area equal to the determined cross-sectional area of the nanochannel.

4. The description of the aldolase molecule and has been added in the Methods – “Biomolecular solutions”.

5. The detailed description of the surface passivation method was added in the Methods – “Supported lipid bilayer”.

6. The detailed description of the preparation steps related to the analysis of the conditioned cell culture medium has been added in the Methods – “Conditioned cell culture medium”.

7. The optical properties of a coated nanochannel differ from the optical properties of a bare nanochannel. The corresponding theory is detailed in the SI section 3 – “Light scattering of a biomolecule inside a coated nanochannel” added to the revised SI.

8. In addition, the implications of this difference to the translation from iOC to MW are discussed at the end of the SI section 4 – “Polarizability of a protein” as follows:

“In this study, the parameter A was determined from SEM images of the nanochannel cross section (Fig. S3). Parameter $\bar{n} = -10.25$ was calculated from Equation 18 and Equation 19, where $n_i = 1.33$ RIU (water), $n_o = 1.46$ RIU (SiO₂) and even representation of TE and TM polarization was assumed, $\bar{n} = 0.5(\bar{n}^{TE} + \bar{n}^{TM})$. For nanochannels coated with a lipid bilayer, a modified parameter $\bar{n}' = \bar{n} \sqrt{I_c / I_c'}$ was used (Equation 26) where $I_c / I_c' = 1.7$ for Channel VI was determined from the intensity of scattered light from the nanochannel before and after the deposition of the lipid bilayer.”

9. The section “ML analysis of conditioned cell culture medium” has been added into Methods.

10. In the Abstract and Conclusions, a note regarding the presented analysis of EVs in cell culture medium was added:

“... Furthermore, we demonstrate its applicability to the analysis of a complex biofluid, using conditioned cell culture medium containing extracellular vesicles as an example. ...”

and

“... Moreover, SLB coating on the nanochannel walls prevented nonspecific binding, thereby enabling analysis of both positively and negatively charged proteins. ...”

and

“... As a second aspect, we have analyzed conditioned cell culture medium collected from human SH-SY5Y cells, which contained a complex mixture of proteins and BNPs. We were able to accurately determine and, importantly, distinguish the size and corresponding distributions of lipoprotein particles and EVs in this setting. Looking forward, this advertises NSM for label-free single-cell studies in real time as it is highly efficient for the analysis of intracellular content or secretomes due to minimized sample dilution in the nanofluidic system. ...”

11. The title of the study was changed to “Nanofluidic Scattering Microscopy for Label-free Weight and Size Screening of Single Diffusing Biological Nanoparticles and Molecules” to reflect the broader scope of the revised manuscript. In addition, couple of minor changes throughout the text were made to present the method in more general and versatile way.

12. Page 19, line 1, N_s is not defined.

Our reply: We have corrected this mistake and added the definition of NS as “... N_s corresponds to spatial averaging (signal was averaged over a diffraction limited spot, N_s=20×20 pixels²) ...”.

Reviewer #2:

The manuscript presents an interferometric scattering imaging modality (Nanofluidic Scattering Microscopy, NSM) in which nanochannels are used to (1) confine the molecules of interest and (2) provide reference light field for interference. In comparison with conventional iSCAT techniques, the authors argue that NSM the major benefits of NSM are: (1) no need for surface attachment, and (2) prolonged sampling time. The authors claim that with these benefits, NSM can achieve 3-fold improvement on molecular weight resolution (30 kDa) comparing with the state-of-the-art iSCAT techniques.

However, based on the theory and data shown, I am not at all convinced that the authors' claims can be supported. To begin with, the authors claim that for iSCAT, “as key point, these three label-free optical single-molecule detection methods require the investigated species to bind to a surface to be ‘visible’” (line 52-54), which is not true: In 2019, Taylor RW et al demonstrated via iSCAT the visualization of microsecond nanoscopic protein motion on a live cell membrane with high precision localization, in

which the focal plane was placed “at a distance of several micrometres above the glass interface” (Nature Photonics 13.7 (2019): 480-487). In the manuscript, the authors confine the molecules of interest using nanochannels of different cross sections (Channel I: 100 nm × 27 nm, Channel II: 100 nm × 72 nm, Channel III: 100 nm × 15 nm, Channel IV: 145 nm × 27 nm, Fig. S3). The largest channel height is only 72 nm whereas the smallest channel height can be as small as 15 nm. In comparison, Taylor RW et al demonstrated that iSCAT can achieve an axial localization precision of 4-6 nm for particles within ~100 nm axial displacement via PSF fitting. In other words, the NSM technique actually need to bring the molecules closer to the surface than conventional iSCAT techniques.

Our reply: While the Reviewer is correct that in the study by Taylor et al. (Nature Photonics 13.7 (2019): 480-487) the focal plane was placed several micrometers above the glass surface, we want to clearly point out that the detection of the “freely moving proteins” in this study is not label free. In fact, this work involves the labelling of the proteins with 20 nm gold nanoparticles (GNPs), which in essence is no different from using other well-established labels like fluorescent tags. Therefore, it is in our opinion not correct to compare this work with our NSM method that is truly label free.

Furthermore, we argue that what the authors in Taylor et al. detect are not “freely moving proteins” but rather protein-GNP complexes where the GNP-tags significantly influence the rates of diffusion due to their important additional volume and mass. In other words, extracting the information about the hydrodynamic radius is impossible in this case. Similarly, we note that GNPs provide an about-one-order-of-magnitude-higher optical contrast than biomolecules of the same size due to their significantly higher refractive index, which is why the authors use them to “enable to visualization of virtually invisible objects”. However, as the key point here, they do not visualize the biomolecules themselves in this way but rather the GNP — just like any other label-based method. Therefore, we are still convinced that the main achievement of NSM is the ability to directly visualize biomolecules that diffuse in a liquid environment, without the use of any labels or any attachment to any surface, and that our key claim is correct and has not been demonstrated before.

As a further aspect related to the Reviewer’s statement “In other words, the NSM technique actually need to bring the molecules closer to the surface than conventional iSCAT techniques”, we note that there is an important difference between being close to a surface and to be chemically bound to a surface since the latter potentially (significantly) alters molecular conformation and function. So, even if we bring the biomolecules close to a surface inside a nanochannel, the key difference is still that we do not require any chemical/electrostatic interaction between the molecule and any kind of surface to be able to visualize the molecules. This is the direct consequence of the fact that the nanochannels ensure that the imaged molecules stay within the focal plane, even if they are freely moving. On the other hand, iSCAT has reported label-free detection of moving proteins only for cases where molecules were

bound to some surface, e.g., myosin 5a on actin filaments (Nano Lett 14, 2065-2070 (2014)) or proteins on clean substrates (Nano Lett 17, 1277-1281 (2017)).

There are additional limitations and concerns by utilizing the scattered light from nanochannels as the reference beam:

1. Clogging of molecules within the channel: the authors selectively choose negatively charged molecules in the study to prevent binding events on the channel walls. This greatly limits the practicality of NSM since in most study the molecules are not all negatively charged.

Our reply: In order to demonstrate the versatility and general practicality of NSM, as discussed in detail above in response to Reviewer #1, we have now added an experimental demonstration of imaging and analysis of the positively charged protein aldolase. To prevent these molecules from binding to the surface, the nanochannel walls were passivated by a supported lipid bilayer (SLB). In addition, also using SLB coated nanochannels, we also now demonstrate analysis of a relevant biofluid, i.e., conditioned cell culture medium containing extracellular vesicles (EVs). This experiment not only convincingly demonstrates the direct applicability of NSM to the study of biofluids but also that clogging is not a major issue if appropriate surface chemistry is applied.

Details concerning the analysis of the EVs in conditioned cell culture medium and of the positively charged aldolase can be found in our response to the 11th comment of Reviewer #1 above or in corresponding new sections added to the main text: "Surface passivation by lipid bilayer", "Analysis of extracellular vesicles in conditioned cell culture medium"; added sections in the Methods: "Supported lipid bilayer", "Conditioned cell culture medium", and added SI section 3 – "Light scattering of a biomolecule inside a coated nanochannel" and SI section 5 – "Error estimate of size determination due to partial-slip boundary condition".

2. Difficulty in adjusting the optical contrast (iOC in the study) in interferometric scattering modality: $iOC = n \cdot \alpha_m / A$. To adjust iOC for small molecules, the only approach in the NSM method is to adjust A, which is the cross-sectional area of the nanochannel in use. In other words, the users would have to redesign and fabricate a customized nanochannel which involves e-beam lithography and thermal oxidation. In iSCAT, this can be simply achieved by swapping the gold dot attenuator of different extinction ratio.

Our reply: The Reviewer is correct that different types of nanofluidics devices would be used to study biological entities with different dimensions/iOC, where A is adjusted for the purpose at hand. While this at first may appear to be a limiting factor, we would argue that it is not since numerous very-well-established experimental techniques — with Scanning Tunneling Microscopy (STM) and Atomic Force Microscopy (AFM) maybe being the most prominent ones — actually require regular exchange of a key component. Similarly, widely used bioanalytical tools like SPR or QCM-D require the regular exchange of the used sensors when new experiments (on different analytes) are targeted. Furthermore, today the fabrication of nanofluidic devices is a very well-established technology. Quoting from L. Bocquet: Nanofluidics coming of age, *Science Materials*, 19, 254-256 (2020): "... The fabrication of nanofluidic devices amenable to systematic investigations was indeed a challenging prerequisite hindering the development of the field. But the domain has since undergone a quantum leap and a general impression from recent papers and conferences is that an exciting period starts for nanofluidics... ... systems that seemed a distant dream ten years ago recently became a reality. It is now possible to fabricate individual artificial channels with nanometric and even sub-nanometric size with manifold geometries."

In addition, a single nanofluidic chip can contain series of nanochannels with varying sizes that could be used in parallel and/or for different experiments. To clarify this, following comment has been added to the Conclusions:

"... Furthermore, rapid progress in detector technologies and machine-learning-based data treatment promises to push the limits of NMS towards even smaller molecules and to enable up to two orders of magnitude increased throughput via parallel analysis of a large number of nanochannels. ..."

Regarding the second aspect of the Reviewer's comment "In iSCAT, this can be simply achieved by swapping the gold dot attenuator of different extinction ratio", we are not entirely clear about what the Reviewer means with "gold dot attenuator" but we assume that s/he probably refers to labelling a molecule with a GNP, as in the work by Tailor et al. already invoked in the comment above (*Nature Photonics* 13.7 (2019): 480-487). While then the statement by the Reviewer is technically correct in terms of changing optical contrast by swapping the attached type of GNP, we again highlight that this strategy in general cannot be considered "label-free" and therefore it is not entirely correct to compare it with NSM. Furthermore, we argue that "simply swapping gold dot attenuator" may not always be as "simple" as it may appear since it every time involves an additional sample-preparation step which by itself may alter the sample. Therefore, in our opinion, it is not necessarily a more efficient solution than

using another NSM chip, in particular once they are available “off-the-shelf”, like the GNPs used for labelling or SPR and QCM sensors.

Furthermore, it is also unclear, as already discussed in the comment above, how iSCAT would ensure the localization of a freely diffusing imaged object within the microscope focal plane since in the aforementioned paper the diffusing object (i.e., the GNP-protein complex) is restricted by the presence of the cell membrane and therefore locked near the focal plane. However, this concept cannot be considered as a general approach for imaging of freely diffusing objects, as the Reviewer is doing here.

3. Limited applications. NSM cannot be used for interferometric scattering imaging in 2D or 3D (for example, cell surface imaging), either can NSM be used for freely diffusive molecule tracking (due to the hindrance effect from the restricted volume of the nanochannel) which greatly limits the scope of the NSM technique as well the throughput (since the molecules are required to flow through a nanochannel rather than freely suspended in solution).

Our reply: As a first general response to this comment, we would like to note that the critique raised here is of technical, rather than fundamental nature. In other words, with the appropriate technical development and refinement, as naturally happening with any newly invented technique over time, many of the current issues that seem to limit applications will be overcome. Simultaneously, we also want to state that it is not realistic to expect that all these limitations are explicitly resolved upon first reporting a new experimental technique, and that so also was not the case with iSCAT that the Reviewer invokes as the key benchmark.

As to a more specific response, we agree with the Reviewer that it might not be possible to use NSM for cell surface imaging but to call this a severe limitation is somewhat exaggerated because, if we again take iSCAT as an example, it is clear that these kinds of experiments only constitute a subgroup of applications. When it comes to molecular tracking of free diffusion, the potential impact of the “hindrance effect” and implications of nanofluidics for the throughput of NSM, we refer to a similar discussion in response to Reviewer #1, which we reproduce here for convenience:

The diffusivity of a molecule inside a nanochannel can indeed be different from that in bulk. In our analysis, we account for surface effects by introducing a hindrance factor K in Eq. 2, whose magnitude is estimated using the phenomenological model suggested by Dechadilok et al. 9. Specifically, we

demonstrate this behavior in Fig. 3F by plotting the theoretical dependency of diffusivity on both molecule and nanochannel size. To clarify this and explain in more detail the effects that our analysis accounts for, we have modified the manuscript in the following way:

“... by approximating it as a hard neutral sphere, its hydrodynamic (Stokes) radius, R_s , can be estimated using the Stokes-Einstein equation corrected for hindrance effects associated with the diffusion of small objects in a restricted volume as⁹

$$R_s = K \cdot (k_B T) / 6\pi\eta D \quad \text{Equation 2}$$

where k_B is the Boltzmann constant, T is temperature, η is the viscosity of the liquid in the nanochannel, and K is the hindrance factor that takes particle-wall hydrodynamic interactions and steric restrictions inside a nanochannel into account. It is dependent on the size of the nanochannel relative to the dimensions of the biomolecule, and can be estimated using a phenomenological model suggested by Dechadilok et al. ⁹ as ... “

As a second aspect, we agree that the diffusivity of a biomolecule inside the nanochannel could be slightly affected by other factors that are not included in the used theory. To clarify this point we have added a brief discussion to the revised manuscript:

“We note here that the diffusivity also can be slightly affected by other surface-related effects that are not included here, such as the partial-slip boundary condition¹⁰.”

Importantly however, in the case of protein analysis, we do not observe any significant contribution from such potential effects in our data, as demonstrated by the fact that our estimated hydrodynamic radii correspond very well with the values measured using electrophoresis in the bulk (Fig. 3F). We have added the following note to the text:

“We find that the theoretically predicted values reproduce the experimentally measured D values well, which suggests that potential surface-related effects are negligible.”

In addition, in the case of the added analysis of extracellular vesicles in conditioned cell culture medium, we have estimated the contribution of the partial-slip boundary condition to be less than 4%, based on published theory¹⁰. The details of the calculation are summarized in the new SI Section 5 – “Error estimate of size determination due to partial-slip boundary condition”.

As a further aspect of our reply to this comment, we want to highlight that the use of a nanofluidic solution not only comes with potential limitations, as stated by the Reviewer, but, as we point out in the Conclusions, it actually also opens up new possibilities that are unique to nanofluidics and that are the reason for the wide and steadily growing application of nanofluidic solutions in biology, biochemistry and biophysics, which include the imaging of DNA (Lab Chip 17, 579-590 (2017)), the analysis of exosomes (Int J Mol Sci 18(6),1153, (2017)), the analysis of the intracellular content of a single cell (Analyst 142, 1689-1696 (2017)), and the analysis of the cell secretion (Small 10, 1514-1522 (2014)). In particular in the last example, the limited volume of a nanochannel even constitutes a critical benefit since it prevents the excessive dilution of the secretome of a single cell and significantly improves mass-transport characteristics, thereby substantially increasing the probability of detection of secreted molecules compared to detection in traditional single-cell microfluidics.

4. Additional costs for each imaging experiment by the nanochannel chips.

Our reply: As also commented above, there are many widely established techniques that heavily rely on consumables, without that limiting their applicability. More technically speaking, the cost for nanochannel chips is much smaller than what one might expect. In fact, we produce today such chips for our in-house use at a cost of less than 10 €/piece including the labor cost of the nanofabrication specialist who makes them, all the user fees of the nanofabrication lab, etc. This is negligible compared to the main cost associated with any research project. Furthermore, as with any manufacturing, cost scales with volume and hence, as the technique will become more established and potentially commercially available, chips are consumables available at the same or even lower cost than SPR sensors or QCM crystals, for example. Hence, the cost of chips most certainly is not a limitation of NSM.

Additionally, the authors' claim on "3-fold improvement on molecular weight resolution" was based on a publication in 2019 (Science 360.6387 (2018): 423-427). However, for BSA the MW resolution is approximately the same. In addition, the current state-of-the-art iSCAT technique can achieve the same if not smaller resolution, for example, recently tubulin monomers of ~ 50 kDa was detected with a bandwidth of 5 kDa (Journal of molecular biology 432.23 (2020): 6168-6172.). BSA is of similar MW in the manuscript, yet the resolution reported using NSM is 20 kDa, which is much larger than the current state-of-the-art method. To claim a 3-fold improvement on molecular weight resolution, the authors need to compare with more than just one sample. The authors claim that the improvement was attributed to the prolonged sampling time. Yet I am suspecting that, since NSM uses polychromatic light, the speckle patterns commonly observed in iSCAT are essentially averaged out, thus significantly reduces the background noise.

Our reply: We would like to clarify that in the iSCAT study describing the detection of tubulin (Journal of molecular biology 432.23 (2020): 6168-6172.) the authors do not state the achieved resolution of iSCAT (defined as full width at half-maximum of the measured peak). The value 5 kDa was used in the publication in the context of the bandwidth of kernel density estimates — a smoothing parameter that has been chosen by the authors to present the data. From the presented images showing the distribution of the measured mass (Figure 1e), we can only estimate the width of the peaks to be approximately 20 kDa — comparable to the resolution reported in (Science 360.6387 (2018): 423-427) for similar range of molecular weights.

However, we agree that our claim “3-fold improvement on molecular weight resolution” is not accurate — the stated improvement was achieved for a molecule in the range of hundreds of kDa, in the range of tens of kDa the resolution is rather comparable to the state-of-art iSCAT. Therefore, we have modified this statement into “up to 3-fold improvement on molecular weight resolution”.

Regarding the reviewer’s comment “Yet I am suspecting that, since NSM uses polychromatic light, the speckle patterns commonly observed in iSCAT are essentially averaged out, thus significantly reduces the background noise.”, in publications (Science 360.6387 (2018): 423-427) or similar iSCAT study (Nat Commun 5, 4495 (2014)) where monochromatic light was used, the authors claim that the iSCAT resolution is limited by photon shot noise, therefore, not influenced by speckle patterns.

Minor critics:

Line 81-82: “In this arrangement, the nanochannels improve the optical contrast of the imaged biomolecules by several orders of magnitude”. What technique are the authors comparing with here with regards to improvement on contrast? Through what mechanism was the contrast improved? Small channel scattering cross section thus small reference field?

Our reply: This statement refers to the intensity of the light scattered by a biomolecule alone, diffusing in bulk solution, outside the nanochannel. This statement is then supported and explained by the paragraph below this statement and Fig. 1 B-D. To make this clear and avoid any confusion, we have rearranged the corresponding text in following way:

“... In this arrangement, the nanochannels ensure the localization of the nano-objects within the microscope focal plane throughout the entire imaging process, similar to other tether-free microscopy methods¹⁹. Most importantly, however, the nanochannels improve the optical contrast of the imaged nano-object by several orders of magnitude. To introduce the underlying principle, we consider a single biomolecule diffusing inside a nanochannel (Fig. 1A). The biomolecule and the nanochannel scatter light coherently into the collection optics, ...”

Reviewer #3:

The authors build on their expertise of conducting biological analysis inside nanofluidic channels to present a new imaging method termed Nanofluidic Scattering Microscopy (NSM). I fully agree with the rationale for the method which does not require surface affinity - this difference sets it firmly apart from other existing methods, most notably iScat, and makes the method very powerful. In this regard, the technique is very elegant and I particularly like the fact that it collects more information, namely a) optical contrast AND hydrodynamic radius and b) it tracks molecules in free motion, so can collect information longer than a comparable surface affinity technique.

Overall, this is a great paper and I congratulate the authors on their excellent work!

Our reply: We would like to thank the Reviewer for this very positive assessment of our work.

I only have 2 minor questions that I would like the authors to address; a) What determines the optimum channel size? I would appreciate a discussion of method performance vs channel size. Related, are there any issues with clogging?

Our reply: We thank the reviewer for a valuable comment since optimization of the channel size is indeed an important topic that has only briefly been mentioned in the context of the obtained resolution for Channel I and II in SI section 8 – “Resolution in molecular weight and hydrodynamic radius”. In order to provide a more general picture of the role of nanochannel size, we have now discussed both theoretical limits, as well as experimental observations, including a discussion of clogging issues and how to resolve them.

The following text has been modified in SI section 8 – “Resolution in molecular weight and hydrodynamic radius”:

“...The determined value of iOC can be translated into molecular weight using Equation 1 in the main text. The resolution in MW, w_{MW} , is then linearly dependent on the resolution in iOC and linearly increases with the cross-sectional area of the nanochannel, $w_{MW}=w_{iOC}\cdot A/(\bar{n}\alpha_{MW})$. We note here that these predictions assume that the shot noise level remains the same, i.e., the intensity and temporal averaging remain the same. In other words, downsizing of the nanochannel cross section improves the optical performance, but only to the level where the intensity of the scattered light saturates the camera at its maximal frame rate. In addition, downsizing the nanochannel cross section presents a challenge for nanofabrication, sets limits on the maximal size of molecules that can enter the channel and be analyzed, and increases the risk of clogging. However, in the section “Surface passivation by lipid bilayer” we show that the risk for clogging can be minimized by surface modification of the nanochannel walls by, e.g., a lipid bilayer, to avoid adsorption of molecules on the surface. In addition, in the section “Analysis of extracellular vesicles in conditioned cell culture medium” we show that the nanochannel dimensions can be tailored to accommodate even BNPs, such as EVs, without any obvious problems related to clogging. Looking forward, an analysis covering a wide range of molecular weights could be enabled by a series of nanochannels with varying sizes used in parallel. To prevent the clogging of the smaller nanochannels by larger molecules present in the sample, we propose that on-chip sorting systems²⁰ could be utilized. ...”

And the following comment has been added to the main text:

“...The positions of the main peaks that correspond to molecular monomers are marked by ellipses in Fig. 3A, B, whose centers correspond to the mean values of $\langle iOC \rangle$ and $\langle D \rangle$, and their horizontal and vertical diameters to the resolution in iOC and D , respectively, defined by the full-width-at-half-maximum (FWHM) of the peaks in Fig. 3C, D. For Channel I this translates into MW resolution of 20 – 30 kDa and for Channel II of 30 – 40 kDa (details in SI Section 8), ...”

b) Please discuss the limits of the method in terms of analyte concentration and mixed populations of particles/molecules.

Our reply: We agree that limits in terms of analyte concentration and mixed populations of particles/molecules are important aspects of single molecule detection. In the revised version of the manuscript, we address these in the following way:

We have added a discussion related to the limits of the method in terms of analyte concentration to the main text at the end of the section “Molecular weight and hydrodynamic radius determination”:

“As final comment, we note that the 28 nM molecular concentration chosen for the experiments is high enough to ensure sufficient throughput and at the same time low enough to enable correct and precise discrimination of individual biomolecules. It corresponds to 0.7 and 2 biomolecules on average per field of view in Channel I and II, respectively. Lower concentrations can be studied by applying a constant flow rather than relying on diffusion alone to improve throughput. For higher concentrations, more advanced particle trackers are in development.”

Regarding the limits in terms of mixed populations, the presented data clearly shows that different populations of the molecules can be detected, e.g., population of monomers and dimers of the proteins (Fig. 3A-D), or subpopulations of ferritin containing different amount of iron (SI section 9, Fig. S23).

In addition, in order to quantify more precisely the ability to resolve different populations, we have added following discussion to the main text (section “A single biomolecule library”):

“...For Channel I this translates into MW resolution of 20 – 30 kDa and for Channel II of 30 – 40 kDa (details in SI Section 8), which defines the limits for resolving different populations in a sample. ...”

Additional implemented changes during the revision:

Besides the changes requested by the reviewers, several other modifications have been made to improve the manuscript:

1. The ML algorithm has been improved — results reported in Fig. 3C (inset) and Fig S4 have therefore slightly changed. Apart from several minor improvements, the performance is considerably boosted mainly by three additions: train-validation on simulated data atop experimental noise, ensemble

modelling, and curriculum learning. Comments explaining these additions have been added to SI Machine Learning (ML) Analysis, in particular:

“The model is also train-validated every 120 epochs (≈ 30000 simulated kymographs) against 150 simulated kymographs (of size 128×512 , 128×1024 , 128×2048) with experimentally measured channel noise, using an 80-20 train-validation split.”

“To train the intensity- and diffusivity-calculating FCNN models, we employed a curriculum learning scheme with intermittent checkpoints to be used for later ensemble modelling prediction. Specifically, the intensity-calculating model was initially trained only on a narrow range of high iOC trajectories, representing the highest SNR and in principle easiest case for the model to begin learning correlations, and then slowly curriculum-learned down to the lowest range of relevant iOC values. In each narrow range of iOC values, checkpoint models which are more accurate in that particular narrow range of values are saved separately. Upon model inference, an initial prediction is made with a model trained on the entire range of iOC values with the scheme described above, and then a second model trained on the narrower range of values makes a second prediction on the same trajectory to achieve higher accuracy. The process is equivalent for diffusivity, with the difference being that the range of D values being trained on increases rather than decreases during curriculum learning.”

2. We have selected different example of a kymograph capturing a rare event of molecular binding to the nanochannel wall (Fig. S5). Compared to the previously selected kymograph, it captures longer trajectory before and after the binding event and therefore better illustrates the typical stochastic behavior of Brownian motion.

Fig. S5. Ferritin–nanochannel wall interaction. Kymograph of a single ferritin molecule inside Channel II. For each time frame, the dark spot in the image corresponds to the position of the ferritin that resides inside the field of the view. The position of the biomolecule changes stochastically in the left and right part of the image, whereas in the central part, it remains fixed for about 0.3 s. These two different behaviors correspond to two different states of a biomolecule – freely diffusing and bound to the nanochannel wall, respectively. The trajectory of the biomolecule was found using a particle tracking algorithm and from the statistics of the movement, the part of the trajectory corresponding to the bound state was identified and excluded from the further analysis (more details in section Particle tracking algorithm).

3. The following co-authors have been added to list of authors: Quentin Lubart, Daniel van Leeuwen, Elin K. Esbjörner who provided the cells cultures and corresponding media used for the newly added EV analysis; David Albinsson who fabricated the nanofluidic chips used for the analysis of positively charged proteins and the cell culture medium containing EVs.

4. Since the analysis of the cell culture medium containing EVs that was added in the re-vised manuscript based on the request of the reviewers constitutes a considerable extension of the main text, the section “The ferritin system” has been moved to SI section 9 to adhere to the length guidelines.

5. Several minor typos and errors have been corrected. All the correction are marked in the text by green color.

References

- 1 Vollmer, F. & Yang, L. Label-free detection with high-Q microcavities: a review of biosensing mechanisms for integrated devices. *Nanophotonics-Berlin* 1, 267-291, doi:10.1515/nanoph-2012-0021 (2012).
- 2 Zhang, P. et al. Plasmonic scattering imaging of single proteins and binding kinetics. *Nat Methods* 17, 1010-1017, doi:10.1038/s41592-020-0947-0 (2020).
- 3 Taylor, A. B. & Zijlstra, P. Single-Molecule Plasmon Sensing: Current Status and Future Prospects. *AcS Sensors* 2, 1103-1122, doi:10.1021/acssensors.7b00382 (2017).

- 4 Piliarik, M. & Sandoghdar, V. Direct optical sensing of single unlabelled proteins and super-resolution imaging of their binding sites. *Nat Commun* 5, 4495, doi:10.1038/ncomms5495 (2014).
- 5 McDonald, M. P. et al. Visualizing Single-Cell Secretion Dynamics with Single-Protein Sensitivity. *Nano Lett* 18, 513-519, doi:10.1021/acs.nanolett.7b04494 (2018).
- 6 Liebel, M., Hugall, J. T. & van Hulst, N. F. Ultrasensitive Label-Free Nanosensing and High-Speed Tracking of Single Proteins. *Nano Lett* 17, 1277-1281, doi:10.1021/acs.nanolett.6b05040 (2017).
- 7 Ortega Arroyo, J. et al. Label-free, all-optical detection, imaging, and tracking of a single protein. *Nano Lett* 14, 2065-2070, doi:10.1021/nl500234t (2014).
- 8 Young, G. et al. Quantitative mass imaging of single biological macromolecules. *Science* 360, 423-427, doi:10.1126/science.aar5839 (2018).
- 9 Dechadilok, P. & Deen, W. M. Hindrance factors for diffusion and convection in pores. *Ind Eng Chem Res* 45, 6953-6959, doi:10.1021/ie051387n (2006).
- 10 Olsen, E. et al. Diffusion of Lipid Nanovesicles Bound to a Lipid Membrane Is Associated with the Partial-Slip Boundary Condition. *Nano Lett* 21, 8503-8509, doi:10.1021/acs.nanolett.1c02092 (2021).
- 11 Vestergaard, C. L., Blainey, P. C. & Flyvbjerg, H. Optimal estimation of diffusion coefficients from single-particle trajectories. *Phys Rev E* 89, doi:ARTN 022726
10.1103/PhysRevE.89.022726 (2014).
- 12 Mawatari, K., Kazoe, Y., Shimizu, H., Pihosh, Y. & Kitamori, T. Extended-nanofluidics: fundamental technologies, unique liquid properties, and application in chemical and bio analysis methods and devices. *Anal Chem* 86, 4068-4077, doi:10.1021/ac4026303 (2014).
- 13 Persson, F. et al. Lipid-based passivation in nanofluidics. *Nano Lett* 12, 2260-2265, doi:10.1021/nl204535h (2012).
- 14 La Verde, V., Dominici, P. & Astegno, A. Determination of Hydrodynamic Radius of Proteins by Size Exclusion Chromatography. *Bio-Protocol* 7, doi:ARTN e2230
10.21769/BioProtoc.2230 (2017).
- 15 van der Pol, E., Boing, A. N., Harrison, P., Sturk, A. & Nieuwland, R. Classification, functions, and clinical relevance of extracellular vesicles. *Pharmacol Rev* 64, 676-705, doi:10.1124/pr.112.005983 (2012).
- 16 Simpson, R. J., Lim, J. W., Moritz, R. L. & Mathivanan, S. Exosomes: proteomic insights and diagnostic potential. *Expert Rev Proteomics* 6, 267-283, doi:10.1586/epr.09.17 (2009).

- 17 Gurunathan, S., Kang, M. H., Jeyaraj, M., Qasim, M. & Kim, J. H. Review of the Isolation, Characterization, Biological Function, and Multifarious Therapeutic Approaches of Exosomes. *Cells* 8, doi:10.3390/cells8040307 (2019).
- 18 Satyanarayana, U. *Biochemistry* (2nd ed.). (Kolkata, India: Books and Allied, 2002).
- 19 Tyagi, S. et al. Continuous throughput and long-term observation of single-molecule FRET without immobilization. *Nat Methods* 11, 297-300, doi:10.1038/nmeth.2809 (2014).
- 20 Bayareh, M. An updated review on particle separation in passive microfluidic devices. *Chemical Engineering & Processing: Process Intensification* 153, 107984 (2020).

Decision Letter, second revision:

Dear Christoph,

Your Article, "Nanofluidic Scattering Microscopy for Label-free Weight and Size Determination of Single Diffusing Biological Nanoparticles and Molecules", has now been seen again by three reviewers. As you will see from their comments below, the referees find the work greatly improved on the whole.

However, referee 2 has some remaining concerns that we think are important. We are interested in the possibility of publishing your paper in *Nature Methods*, but would like to consider your response to these concerns before we reach a final decision on publication.

We therefore invite you to revise your manuscript to address these concerns. We do not expect you to overcome novelty issues (point 1), but please do cite/discuss these papers in your discussion.

Regarding point 2, we do ask that you use a standard experiment to confirm your exosome observations. '

For point 3, we ask that you discuss motion artifacts and how you deal with them.

We also ask that you address the remaining more minor concerns.

* include a point-by-point response to the reviewers and to any editorial suggestions

* please underline/highlight any additions to the text or areas with other significant changes to facilitate review of the revised manuscript

* address the points listed described below to conform to our open science requirements

* ensure it complies with our general format requirements as set out in our guide to authors at www.nature.com/naturemethods

* resubmit all the necessary files electronically by using the link below to access your home page

[Redacted] This URL links to your confidential home page and associated information about manuscripts you may have submitted, or that you are reviewing for us. If you wish to forward this email to co-authors, please delete the link to your homepage.

We hope to receive your revised paper within XX weeks [****ED TO CUSTOMIZE AS NEEDED****]. If you cannot send it within this time, please let us know. In this event, we will still be happy to reconsider your paper at a later date so long as nothing similar has been accepted for publication at Nature Methods or published elsewhere.

OPEN SCIENCE REQUIREMENTS

REPORTING SUMMARY AND EDITORIAL POLICY CHECKLISTS

Please note that these forms are dynamic ‘smart pdfs’ and must therefore be downloaded and completed in Adobe Reader. We will then flatten them for ease of use by the reviewers. If you would like to reference the guidance text as you complete the template, please access these flattened versions at <http://www.nature.com/authors/policies/availability.html>.

IMAGE INTEGRITY

DATA AVAILABILITY

All novel DNA and RNA sequencing data, protein sequences, genetic polymorphisms, linked genotype and phenotype data, gene expression data, macromolecular structures, and proteomics data must be deposited in a publicly accessible database, and accession codes and associated hyperlinks must be provided in the “Data Availability” section.

To further increase transparency, we encourage you to provide, in tabular form, the data underlying the graphical representations used in your figures. This is in addition to our data-deposition policy for specific types of experiments and large datasets. For readers, the source data will be made accessible directly from the figure legend. Spreadsheets can be submitted in .xls, .xlsx or .csv formats. Only one (1) file per figure is permitted: thus if there is a multi-paneled figure the source data for each panel should be clearly labeled in the csv/Excel file; alternately the data for a figure can be included in multiple, clearly labeled sheets in an Excel file. File sizes of up to 30 MB are permitted. When submitting source

data files with your manuscript please select the Source Data file type and use the Title field in the File Description tab to indicate which figure the source data pertains to.

Please include a “Data availability” subsection in the Online Methods. This section should inform readers about the availability of the data used to support the conclusions of your study, including accession codes to public repositories, references to source data that may be published alongside the paper, unique identifiers such as URLs to data repository entries, or data set DOIs, and any other statement about data availability. At a minimum, you should include the following statement: “The data that support the findings of this study are available from the corresponding author upon request”, describing which data is available upon request and mentioning any restrictions on availability. If DOIs are provided, please include these in the Reference list (authors, title, publisher (repository name), identifier, year). For more guidance on how to write this section please see: <http://www.nature.com/authors/policies/data/data-availability-statements-data-citations.pdf>

CODE AVAILABILITY

Please include a “Code Availability” subsection in the Online Methods which details how your custom code is made available. Only in rare cases (where code is not central to the main conclusions of the paper) is the statement “available upon request” allowed (and reasons should be specified).

ORCID

Nature Methods is committed to improving transparency in authorship. As part of our efforts in this direction, we are now requesting that all authors identified as ‘corresponding author’ on published papers create and link their Open Researcher and Contributor Identifier (ORCID) with their account on the Manuscript Tracking System (MTS), prior to acceptance. This applies to primary research papers only. ORCID helps the scientific community achieve unambiguous attribution of all scholarly contributions. You can create and link your ORCID from the home page of the MTS by clicking on ‘Modify my Springer Nature account’. For more information please visit www.springernature.com/orcid.

Sincerely,
Rita

Rita Strack, Ph.D.
Senior Editor
Nature Methods

Reviewers' Comments:

Reviewer #1:

Remarks to the Author:

This major revision has addressed all my previous comments nicely. The addition of detection of EV shows the feasibility of practical application of this new technology. I do not have further questions and I think the manuscript is ready to be accepted.

Reviewer #2:

Remarks to the Author:

In the revised manuscript, the authors included additional results for positively charged proteins and extracellular vesicles characterization. Some of my previous comments were addressed while others require further clarification.

1. My major concern is the central argument that the authors are trying to establish in the article: NSM made a “leapfrogging step in the field” via “the ability to label-free image and track diffusing single biomolecules directly in solution” (Line 57-59), since the existing “label-free optical single-molecule detection methods require the investigated species to bind to a surface to be visible”. In the rebuttal letter, the authors argued that the key difference between NSM and iSCAT is that NSM does not require any chemical/electrostatic interaction between the analyte and the surface while all existing methods do (Line 359-360). However, several articles (Nat. Methods 18.10 (2021): 1247-1252; Nat. Methods 18.10 (2021): 1239-1246) have already demonstrated that iSCAT can be used to track and measure truly free diffusing proteins on supported lipid bilayers without the need to bond with the surface. These two publications have impaired the novelty and technical significance of the proposed method, which claims to address the limit of iSCAT on surface binding requirement.

2. Additional critics on the extracellular vesicles measurement: the authors present no downstream tests to validate the claim that the extracellular vesicles they detect are intact (rather than fragments) and match the predicted size. Specially, considering the wide range of size distribution (40 to 1000 nm in

diameter, according to Ref 36) of extracellular vesicles, it is natural to suspect small cross section of the nanochannel ($225 \times 200 \text{ nm}^2$) will have size screening effect on the detected extracellular vesicle particles. A direct comparison of sizing results obtained by conventional technique (TEM, DLS for example) and by NSM is therefore warranted.

3. The current temporal resolution is 5 ms, during which the molecules under examination could have a mean path of 450-700 nm. One issue that came into question is the effect of motion blurring: within one time step, the analyte travels a mean path of 450-700 nm, which by rough estimation covers tens of pixels, thus the contrast of the analyte within one frame is actually an average of the ground-truth (stationary) target contrast and that of the background signal. Therefore it is expected that dynamic (diffusing) mass measurement will be less sensitive than that of the stationary (surface bound) case. A simple and straightforward investigation is to plot the contrast shown in Fig. S5 as a function of time, and examine the contrast distribution respectively for Brownian-drive motion and surface-bound state.

Minor critics: 1. Multiple references in the manuscript are missing (Line 211, Line 251, etc.). 2. In line 325 the authors claim to have "collected a significant number of trajectories of" extracellular vesicles yet from Fig. 5B at most 16 trajectories of extracellular vesicles were analyzed. Due to the extremely small sample size (comparing with other conventional sizing techniques) even under fluidic flow, I don't think the distribution of extracellular vesicles were accurately portrayed in the result. 3. Can the authors comment on how the number "two orders of magnitude" on line 374 was obtained? I cannot find any evidence/deduction supporting this claim of improvement. 4. Supplemental video has neither temporal stamp nor color bar and scale bar.

Reviewer #3:

Remarks to the Author:

I thank the authors for addressing my concerns.

Author Rebuttal, second revision:

Reviewer #2:

In the revised manuscript, the authors included additional results for positively charged proteins and extracellular vesicles characterization. Some of my previous comments were addressed while others require further clarification.

1. My major concern is the central argument that the authors are trying to establish in the article: NSM made a "leapfrogging step in the field" via "the ability to label-free image and track diffusing single biomolecules directly in solution" (Line 57-59), since the existing "label-free optical single-molecule detection methods require the investigated species to bind to a surface to be visible". In the rebuttal letter, the authors argued that the key difference between NSM and iSCAT is that NSM does not require any chemical/electrostatic interaction between the analyte and the surface while all existing methods do (Line 359-360). However,

several articles (Nat. Methods 18.10 (2021): 1247-1252; Nat. Methods 18.10 (2021): 1239-1246) have already demonstrated that iSCAT can be used to track and measure truly free diffusing proteins on supported lipid bilayers without the need to bond with the surface. These two publications have impaired the novelty and technical significance of the proposed method, which claims to address the limit of iSCAT on surface binding requirement.

Our reply:

We believe that our statement “leapfrogging step in the field” via “the ability to label-free image and track diffusing single biomolecules directly in solution” (Line 57-59) is valid since both publications mentioned by the reviewer (Nat. Methods 18.10 (2021): 1247-1252; Nat. Methods 18.10 (2021): 1239-1246) describe studies of membrane-associated biomolecular processes, where the studied analyte is bound to the supported lipid bilayer, i.e. not freely diffusing in solution.

However, since these publications present significant achievement in the field of membrane-associated proteins and are of high relevance, we have included them in the Introduction section:

“... Label-free single-biomolecule detection has recently been enabled by dielectric microresonators⁹, plasmonic approaches based on metallic continuous films¹⁰ and nanostructures¹¹, and interferometric scattering microscopy (iSCAT)¹²⁻¹⁶. The latter has, e.g., been used to investigate single cell secretion dynamics¹³, protein motion on a dielectric substrate¹⁴, on a supported lipid bilayer^{17,18}, or on actin filaments¹⁵. ...”

Furthermore, we have reworded from “leapfrogging step” to “important step”.

2. Additional critics on the extracellular vesicles measurement: the authors present no downstream tests to validate the claim that the extracellular vesicles they detect are intact (rather than fragments) and match the predicted size. Specially, considering the wide range of size distribution (40 to 1000 nm in diameter, according to Ref 36) of extracellular vesicles, it is natural to suspect small cross section of the nanochannel (225 x 200 nm²) will have size screening effect on the detected extracellular vesicle particles. A direct comparison of sizing results obtained by conventional technique (TEM, DLS for example) and by NSM is therefore warranted.

Our reply: We agree with the reviewer that a comparison with a conventional technique increases credibility of the presented results. Therefore, we now include a size distribution measurement of the conditioned cell culture medium obtained by Nanoparticle Tracking Analysis (NTA) that confirms the size distribution of the extracellular vesicles obtained by NSM.

Regarding the mentioned size screening effect, we note that the larger microvesicles were not present in our sample, as it can be seen in the NTA measurement. That is because we use a 0.22 µm filter in the EV purification, as it is mentioned in Methods – “Conditioned cell culture medium”.

1. The following section containing the NTA measurements and their discussion has been

added to the SI:

“10. Characterization of the conditioned cell culture medium using Nanoparticle Tracking Analysis

In order to validate the results of the NSM analysis of the conditioned cell culture medium, we have carried out a particle size distribution measurement of the same sample using Nanoparticle Tracking Analysis (NTA, Fig. S25), with which we identified a population of extracellular vesicles (EVs) whose mode value of RS = 36.5 nm is in very good agreement with the mode value identified by NSM (RS = 34 nm, Fig. 5D). The absence of the population of smaller particles (lipoproteins) in the NTA data is the consequence of these small particles being below the limit of detection of the method.

Fig. S25. Size distribution of BNPs present in the conditioned cell culture medium characterized by NTA. The population of EVs is marked by the blue arrow.

2. The following reference to this section has been added to the main text of the manuscript: “The identified RS of 10 – 13 nm and 20 – 70 nm (Fig. 5D,E) correspond to values reported for low-density lipoproteins³⁹ and EVs³⁸, respectively. In addition, the obtained size of EVs was validated by a comparative size distribution measurement using Nanoparticle Tracking Analysis (NTA, Fig. S25) (more details in Methods – “NTA”). This clearly demonstrates the applicability of NSM to detect and, importantly, distinguish non-labelled analytes of biological relevance in a complex sample mixture retrieved from cell culture.”

3. Details of the reference method were added to Methods section:

“Nanoparticle Tracking Analysis (NTA)

NTA was performed using a Malvern NanoSight LM10 instrument equipped with a 488 nm laser. It was operated in scattering mode and under a flow rate of 100 (B10 mL/min) obtained with a Nanosight syringe pump module, and with the camera level set to 15. The sample was diluted 1000x in PBS and analysed in a set of five videos of 60 seconds each. The videos were analysed with the built-in NTA 3.2 software using a detection threshold of 5 to be able to determine optimized size distributions and concentrations. The buffer viscosity was considered as that of water at 21°C. Concentration values (particles/ml) were extracted and plotted.”

3. The current temporal resolution is 5 ms, during which the molecules under examination could have a mean path of 450-700 nm. One issue that came into question is the effect of motion blurring: within one time step, the analyte travels a mean path of 450-700 nm, which by rough estimation covers tens of pixels, thus the contrast of the analyte within one frame is actually an average of the ground-truth (stationary) target contrast and that of the background signal. Therefore it is expected that dynamic (diffusing) mass measurement will be less sensitive than that of the stationary (surface bound) case. A simple and straightforward investigation is to plot the contrast shown in Fig. S5 as a function of time, and examine the contrast distribution respectively for Brownian-drive motion and surface-bound state.

Our reply: We thank the reviewer for the valuable comment. The motion blur indeed decreases the contrast of the biomolecule signature. However, the effect on the sensitivity or detection limit of NSM is rather marginal, as we explain in more detail below.

To evaluate the optical signature of a biomolecule, we use the zero intensity moment (m , sum of intensities over a fixed number of pixels) instead of its optical contrast (depth of the intensity dip – see Methods – Particle tracking). This way, the signal-to-noise ratio is higher for both stationary and diffusing biomolecules. Moreover, m is less dependent on the shape of the illuminated spot, i.e., less dependent on slight defocusing or motion blur. We illustrate this on the plot of m in Fig. S5 as a function of time. It can be seen that there is no apparent difference between the values corresponding to the molecule in Brownian motion and in a surface-bound state.

Fig. S5. Ferritin–nanochannel wall interaction. (A) Kymograph of a single ferritin molecule inside Channel II. For each time frame, the dark spot in the image corresponds to the position of the ferritin that resides inside the field of the view. The position of the biomolecule changes stochastically in the left and right part of the image, whereas in the central part, it remains fixed for about 0.3 s. These two different behaviors correspond to two different states of a biomolecule – freely diffusing and bound to the nanochannel wall, respectively. The trajectory of the biomolecule was found using a particle tracking algorithm and from the statistics of the movement, the part of the trajectory corresponding to the bound state was identified and excluded from the further analysis (more details in section Particle tracking algorithm). (B) Zero order intensity moments (m) of minima corresponding to the found trajectory. In addition, we have quantified in detail the effect of motion blur on zero intensity moments for the whole range of studied biomolecules using the simulated response data. It shows that for molecules with higher diffusivity than ferritin, the m values are only 15% lower compared to the molecules with low diffusivity and therefore the effect of motion blur can be considered rather marginal. We have added these results in the SI – “6. Evaluation of data processing”.

“... To match the frame rate of the recorded data (200 frames per second) and to mimic the continuous illumination, every time frame of the simulated response with temporal length of $5 \cdot 10^{-3}$ s was averaged over 100 time-steps ($\Delta t = 5 \cdot 10^{-5}$ s) of the generated biomolecule positions, as

$$I_i^r(x) = \frac{1}{100} \sum_{j=1}^{100} \frac{iOC}{\sqrt{\pi} W_{DLS}} \sum_{p=1}^{20} \exp \left[- \left(\frac{x - x_{100(l-1)+j}^p}{W_{DLS}} \right)^2 \right]$$

where x is a space coordinate that corresponds to the space coordinate of the recorded data. For each combination of values of $iOC_{def} = 10^{-4}, 2 \cdot 10^{-4}, 5 \cdot 10^{-4}, 10^{-3}, 2 \cdot 10^{-3} \mu\text{m}$, and $D_{def} = 10, 20, 50 \mu\text{m}^2/\text{s}$, 10 different responses with a temporal length of 10000 frames were generated (selected examples are shown in Fig. S15). We note that the movement of a biomolecule within one time frame results in motion blur – broadening and shallowing of the intensity dip (insets Fig. S15).

The generated response was then combined with recorded background signal (Fig. S16) as $I = I' \cdot I''$ and kymographs were created according to the procedure described in section Removal of the background (Fig. S17), positions of biomolecules were found using the algorithm described in section Particle tracking algorithm. In the first step of the algorithm, the positions of all minima in the kymograph were found and their zero order intensity moments (m) were calculated. Since the m -values are calculated as a sum of intensities of a fixed fraction of a peak, and since the shape of the peak varies due to motion blur, m -values are also slightly dependent on the diffusivity (Fig. S18). Specifically, the mean value of m corresponding to $D = 50 \mu\text{m}^2/\text{s}$ is 15% lower compared to the mean value of $D = 10 \mu\text{m}^2/\text{s}$. Nevertheless, motion blur has a minimal effect on determination of the iOC value, as it is calculated as integral value of the whole intensity dip that always remains constant.

Fig. S15. Examples of the simulated response of biomolecules defined by different combinations of iOC_{dif} and D_{dif} . The simulated data contained 844, 1036, and 1745 trajectories for data set corresponding to $D_{dif} = 10, 20, 50 \mu\text{m}^2/\text{s}$ with various temporal lengths (N frames). Insets: intensity profiles at 5 selected points in time illustrating motion blur - the broadening and shallowing of the intensity dip for molecules with high D.

Fig. S18. Examples of zero order intensity moment histograms obtained from a selected kymograph of 10000 frame temporal length corresponding to the simulated response of biomolecules defined by $iOC_{def} = 5 \cdot 10^{-4} \mu\text{m}$ and different D_{def} (parts of the selected kymograph are shown on Fig. S17)."

Minor critics:

1. Multiple references in the manuscript are missing (Line 211, Line 251, etc.).

Our reply: We thank the reviewer for pointing this out. It was caused by the wrong translation from Microsoft Word to PDF due to a corrupted dynamic reference. We have now fixed this error.

2. In line 325 the authors claim to have "collected a significant number of trajectories of" extracellular vesicles yet from Fig. 5B at most 16 trajectories of extracellular vesicles were analyzed. Due to the extremely small sample size (comparing with other conventional sizing techniques) even under fluidic flow, I don't think the distribution of extracellular vesicles were accurately portrayed in the result.

Our reply: We agree that this statement can be misleading. In fact, what we meant is that we

have collected a significant number of trajectories of bionanoparticles (BNPs) in general, whereof the larger fraction is from the population of lipoproteins and the smaller fraction from extracellular vesicles.

To clarify this point, we have modified the text in the following way:

“... Using this setup, we then collected a significant number of trajectories of different BNPs, from which we derive iOC and D translated to RS using Equation 2 (the applicability of the no-slip condition is discussed in SI section 5). This analysis reveals two distinct populations (Fig. 5B): one in the small iOC/RS regime (red circle) corresponding to the majority of trajectories, and one in the large iOC/RS regime (blue circle) corresponding to a minority of trajectories. ...”

3. Can the authors comment on how the number "two orders of magnitude" on line 374 was obtained? I cannot find any evidence/deduction supporting this claim of improvement.

Our reply: Our current study involves only the analysis of a single nanochannel. The two orders of magnitude improvement in throughput can be achieved by parallel analysis of hundreds of nanochannels imaged in the same field of view.

To clarify our statement in the text we have modified it in the following way:

“Furthermore, rapid progress in detector technologies and machine-learning-based data treatment promises to push the limits of NSM towards even smaller molecules and to enable up to two orders of magnitude increased throughput via parallel analysis of hundreds of nanochannels that can be fitted in the field of view of a microscope⁴¹.”

4. Supplemental video has neither temporal stamp nor color bar and scale bar.

Our reply: We have added temporal stamp, color bar, and scale bar in the modified Movie 1 that is attached.

Decision Letter, third revision:

Dear Christoph,

Thanks for your recent help with your paper. I need to begin this email by saying that I made a small error while handling your paper. I thought I had sent an accept-in-principle decision on the previous version of your paper rather than a revise decision.

The reason this matters now is that we have some formatting that must be done between accept-in-principle and final accept that you have not gotten the necessary instructions for. We are preparing these for you now, and hope to have them to you within a week. My apologies for this oversight!

Here is your formal accept-in-principle decision that outlines what we'll need from you. Thankfully, we have saved ourselves some time for the line editing stage, which usually comes after this!

Please let me know if you have any questions.

Thank you for submitting your revised manuscript "Label-Free Nanofluidic Scattering Microscopy of Size and Mass of Single Diffusing Molecules and Nanoparticles" (NMETH-A45959C). We have reviewed the updates and are happy in principle to publish it in Nature Methods, pending minor revisions to comply with our editorial and formatting guidelines.

TRANSPARENT PEER REVIEW

Nature Methods offers a transparent peer review option for new original research manuscripts submitted from 17th February 2021. We encourage increased transparency in peer review by publishing the reviewer comments, author rebuttal letters and editorial decision letters if the authors agree. Such peer review material is made available as a supplementary peer review file. Please state in the cover letter 'I wish to participate in transparent peer review' if you want to opt in, or 'I do not wish to participate in transparent peer review' if you don't. Failure to state your preference will result in delays in accepting your manuscript for publication.

Thank you again for your interest in Nature Methods Please do not hesitate to contact me if you have any questions.

Sincerely,
Rita

Rita Strack, Ph.D.
Senior Editor
Nature Methods

ORCID

IMPORTANT: Non-corresponding authors do not have to link their ORCID but are encouraged to do so. Please note that it will not be possible to add/modify ORCID at proof. Thus, please let your co-authors know that if they wish to have their ORCID added to the paper they must follow the procedure described in the following link prior to acceptance:

Final Decision Letter:

Dear Christoph,

I am pleased to inform you that your Article, "Label-Free Nanofluidic Scattering Microscopy of Size and Mass of Single Diffusing Molecules and Nanoparticles", has now been accepted for publication in Nature Methods. Your paper is tentatively scheduled for publication in our June or July print issue, and will be published online prior to that. The received and accepted dates will be April 30, 2021 and April 12, 2022. This note is intended to let you know what to expect from us over the next month or so, and to let you know where to address any further questions.

Your paper will now be copyedited to ensure that it conforms to Nature Methods style. Once proofs are generated, they will be sent to you electronically and you will be asked to send a corrected version within 24 hours. It is extremely important that you let us know now whether you will be difficult to contact over the next month. If this is the case, we ask that you send us the contact information (email, phone and fax) of someone who will be able to check the proofs and deal with any last-minute problems.

If, when you receive your proof, you cannot meet the deadline, please inform us at rjsproduction@springernature.com immediately.

Once your manuscript is typeset and you have completed the appropriate grant of rights, you will receive a link to your electronic proof via email with a request to make any corrections within 48 hours. If, when you receive your proof, you cannot meet this deadline, please inform us at rjsproduction@springernature.com immediately.

Once your paper has been scheduled for online publication, the Nature press office will be in touch to confirm the details.

Content is published online weekly on Mondays and Thursdays, and the embargo is set at 16:00 London time (GMT)/11:00 am US Eastern time (EST) on the day of publication. If you need to know the exact publication date or when the news embargo will be lifted, please contact our press office after you have

submitted your proof corrections. Now is the time to inform your Public Relations or Press Office about your paper, as they might be interested in promoting its publication. This will allow them time to prepare an accurate and satisfactory press release. Include your manuscript tracking number NMETH-A45959D and the name of the journal, which they will need when they contact our office.

About one week before your paper is published online, we shall be distributing a press release to news organizations worldwide, which may include details of your work. We are happy for your institution or funding agency to prepare its own press release, but it must mention the embargo date and Nature Methods. Our Press Office will contact you closer to the time of publication, but if you or your Press Office have any inquiries in the meantime, please contact press@nature.com.

If you are active on Twitter, please e-mail me your and your coauthors' Twitter handles so that we may tag you when the paper is published.

Please note that Nature Methods is a Transformative Journal (TJ). Authors may publish their research with us through the traditional subscription access route or make their paper immediately open access through payment of an article-processing charge (APC). Authors will not be required to make a final decision about access to their article until it has been accepted. Find out more about Transformative Journals

Authors may need to take specific actions to achieve compliance with funder and institutional open access mandates. If your research is supported by a funder that requires immediate open access (e.g. according to Plan S principles) then you should select the gold OA route, and we will direct you to the compliant route where possible. For authors selecting the subscription publication route, the journal's standard licensing terms will need to be accepted, including self-archiving policies. Those licensing terms will supersede any other terms that the author or any third party may assert apply to any version of the manuscript.

To assist our authors in disseminating their research to the broader community, our SharedIt initiative provides you with a unique shareable link that will allow anyone (with or without a subscription) to read the published article. Recipients of the link with a subscription will also be able to download and print the PDF. As soon as your article is published, you will receive an automated email with your shareable link.

Please note that you and your coauthors may order reprints and single copies of the issue containing your article through Nature Research Group's reprint website, which is located at <http://www.nature.com/reprints/author-reprints.html>. If there are any questions about reprints please send an email to author-reprints@nature.com and someone will assist you.

Best regards,
Rita